# I-ASIDE: Towards the Global Interpretability of Image Model Robustness through the Lens of Axiomatic Spectral Importance Decomposition

## Abstract

Robust decisions leverage a high proportion of robust features. Natural images have spectral non-uniformity and the majority of spectral energy concentrates on low-frequency components. A change with an infinitesimal amount of energy on the high-frequency components can rewrite the features dominated by high-frequency components. Image models are parameterized general non-linear signal filters. The spectral structures of the model responses to inputs determines the fragility of the learned feature representations. The spectral importance decomposition of models can thus reflect model robustness in response to feature perturbations. To this end, we formulate the spectral importance decomposition problem, and, present **I**mage **A**xiomatic **S**pectral **I**mportance **D**ecomposition **E**xplanation (**I-ASIDE**) – a model-agnostic global interpretability method – to quantify model global robustness and understand how models respond to perturbations. We theoretically show that **I-ASIDE** decomposes the mutual information between feature representations and labels onto spectrum. Our approach provides a unique insight into interpreting model global robustness from the perspective of information theory and enables a considerable number of applications in research, from understanding model robustness, to studying learning dynamics, to assessing label noise, to investigating adversarial vulnerability, to studying out-of-distribution robustness, etc. We showcase multiple applications to endorse these claims.

## 1 Introduction

Global interpretability summarizes the decision dynamics of neural networks *en masse* in contrast to instance-wise local interpretability (Lipton, 2018; Zhang et al., 2021). Local interpretability for image models has achieved great success (Sundararajan et al., 2017; Smilkov et al., 2017; Linardatos et al., 2020; Selvaraju et al., 2017; Arrieta et al., 2020; Zhou et al., 2016; Ribeiro et al., 2016; Lundberg & Lee, 2017; Lakkaraju et al., 2019; Guidotti et al., 2018; Bach et al., 2015; Montavon et al., 2019; Shrikumar et al., 2017), yet quantifiable global interpretability remains virtually unexplored. A brief literature review regarding global interpretability for image models is provided in Section 3.

The global robustness (Hendrycks & Dietterich, 2019; Bai et al., 2021; Goodfellow et al., 2014; Silva & Najafirad, 2020) of models reflects an intrinsic property of models and delineates a crucial aspect towards interpretability and trustworthiness. To this end, we present **I-ASIDE**[1], a model-agnostic method, to quantify the global robustness of image models and understand how models respond to perturbations. Unlike prior works, **I-ASIDE** directly quantifies model global robustness and enables a considerable number of applications in deep learning research across multiple domains (See Section 5). We also theoreticaly show that **I-ASIDE** *de facto* decomposes the mutual information between feature representations and labels. This can provide profound insights into understanding how features are represented inside black box models. We will provide five application showcases to demonstrate the potential applications of **I-ASIDE** framework.

---

[1]Anonymized reproducibility: `https://anonymous.4open.science/r/IASIDE_reproducibility-F8BC/`.

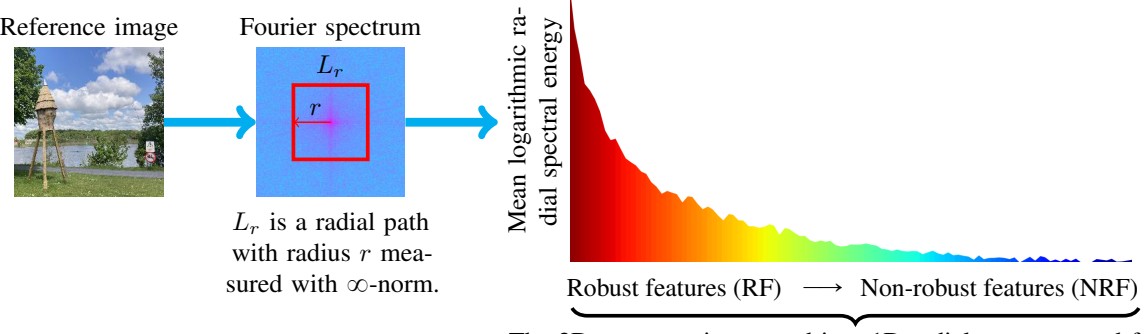

The 2D spectrum is mapped into 1D radial spectrum and further divided into $M$ bands which are denoted from $\mathcal{I}_0$ to $\mathcal{I}_{M-1}$.

Figure 1: This diagram illustrates the correlation between the power-law like radial spectral energy density of natural images and feature robustness. According to the Parseval identity theorem, small spatial perturbations targeting on high-frequency components can rewrite the features dominated by high-frequency components.

The term use of 'feature' is ambiguous and nuanced in deep learning community. In this research, we clarify and differ the term into: (1) Input features (pixel features) and the representations of input features (feature representations). The pixels of images are referred as 'input features' or 'pixel features'. The 'input features' are fed into image models and the models represent the features inside. The way that models represent the input features is referred as 'feature representations'. Empirically, we can 'observe' how models represent the input features by looking into the outputted probability distributions.

Feature representations are the representations of input features that models learn to represent relevant information while neglect irrelevant information (Bengio et al., 2013; Goodfellow et al., 2016). The feature representation robustness largely delimits decision robustness afterwards. Robust decisions leverage more robust feature representations while non-robust decisions leverage less robust representations. The correlation between the spectral energy density distributions of input features and the spectral structures of the responses of models to inputs plays a crucial aspect in feature representation robustness. Consequently, the signal spectrum can index the robustness of features: Low-frequency indicates robust features while high-frequency indicates non-robust features.

Natural images have spectral non-uniformity and the majority of spectral energy concentrates on low-frequency components (See Figure 1 and Figure 2(a)). According to the Parseval identity theorem – the energy of the perturbations in the spatial domain and the spectral domain is conservative and equivalent, the small spatial perturbations targeting on high-frequency components can rewrite those feature representations dominated by high-frequency components. Figure 2 shows four example spectrum from adversarial perturbations to out-of-distribution (OOD) perturbations. The small spatial perturbations in both adversarial noises and OOD noises have considerable energy concentrating on high-frequency components and can have significant impacts on the features dominated by high-frequent components.

Ilyas et al. use the terms 'robust feature (RF)' and 'non-robust feature (NRF)' to theoretically analyze the adversarial robustness problem from the perspective of feature representations and argue that the presence of non-robust features can incur model robustness issues (Ilyas et al., 2019; Tsipras et al., 2018). We adopt the use of their terms and step forward to rigorously discuss the robust decision problem in a broader scope beyond adversarial robustness.

Measuring the ratio of the robust features in decisions can help to interpret model inference dynamics globally from the perspective of feature representations. Unfortunately, discriminating features between being robust and being non-robust is difficult. We have noticed the correlation between the power-law like spectral structures of natural images and feature robustness. Feature robustness can thus be indexed by

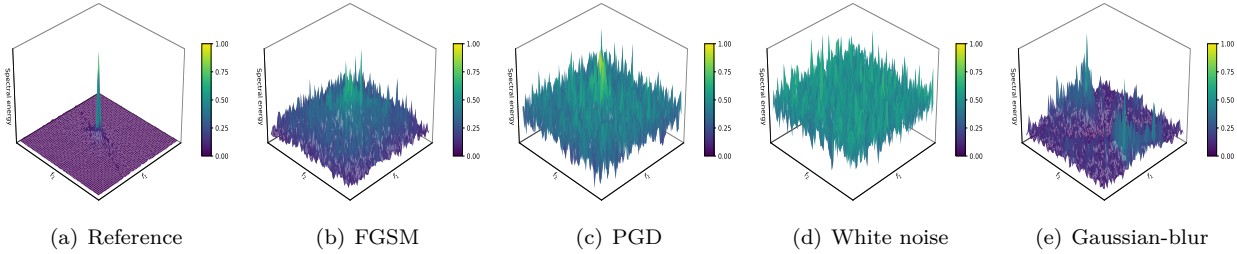

(a) Reference      (b) FGSM      (c) PGD      (d) White noise      (e) Gaussian-blur

Figure 2: This shows the spectrum examples from four perturbation sources: (b) Adversarial FGSM perturbation with $eps = 0.1$, (c) adversarial PGD perturbation with $eps = 0.1$, (d) OOD white noise with $\sigma = 0.1$ and (e) OOD blurring with Gaussian kernel $3 \times 3$. The (a) is the reference spectrum of the reference image (See Figure 1). The adversarial samples are obtained by using a *resnet50* pre-trained on *ImageNet*. The colorbars in the figures indicate frequency-domain magnitudes which are normalized into $[0, 1]$. The perturbation noises are computed by $\Delta \boldsymbol{x} := \boldsymbol{x}^* - \boldsymbol{x}$ where $\boldsymbol{x}$ is some clean sample and $\boldsymbol{x}^*$ is the corresponding perturbed sample. We apply Fourier analysis on the min-max normalized $\Delta \boldsymbol{x}$ and show the results. The normalization does not affect the results from Fourier analysis since Fourier operator is a linear operator. The examples show that several usual adversarial perturbations and OOD perturbations have considerable spectral energy concentrating on high-frequency components. The perturbations targeting on high-frequency components can rewrite the features dominated high-frequency components.

radial spectrum (See Figure 1). We refer the radial spectrum as the 'robustness spectrum' hereafter. The decompositions of the mutual information of the feature representations of inputs and labels with respect to the robustness spectrum can summarize model decision robustness, and quantitatively reflect how neural networks respond to signals in the frequency-domain.

## 2 Contributions

We aim to propose a method of measuring the global spectral importance of image models which is the major focus. The spectral decompositions are a unique solution satisfying a set of desirable axioms. We formulate this problem by using Shapley value theory (Roth, 1988; Aumann & Shapley, 2015). Our approach provides a unique insight into interpreting the global decision robustness of image models and enables multiple applications in the deep learning research community.

We claim our major contributions as:

- We introduce a method of measuring the spectral importance of image models;

- We theoretically justify the proposed method from the perspective of information theory and coalition game theory;

- We demonstrate the potential of our **I-ASIDE** by showcasing multiple applications in a variety of research fields.

The application showcases are:

- Quantifying model global robustness by examining spectral importance distributions (See the experiments regarding spectral importance distributions in Figure 4, summarized numerical comparison and t-SNE projection in Figure 8);

- Understanding the learning behaviours of image models on the datasets with supervision noise (See the experiments in Figure 9);

- Investigating the learning dynamics of models from the perspective of the evolution of feature representation robustness in optimization (See the experiments in Figure 10);

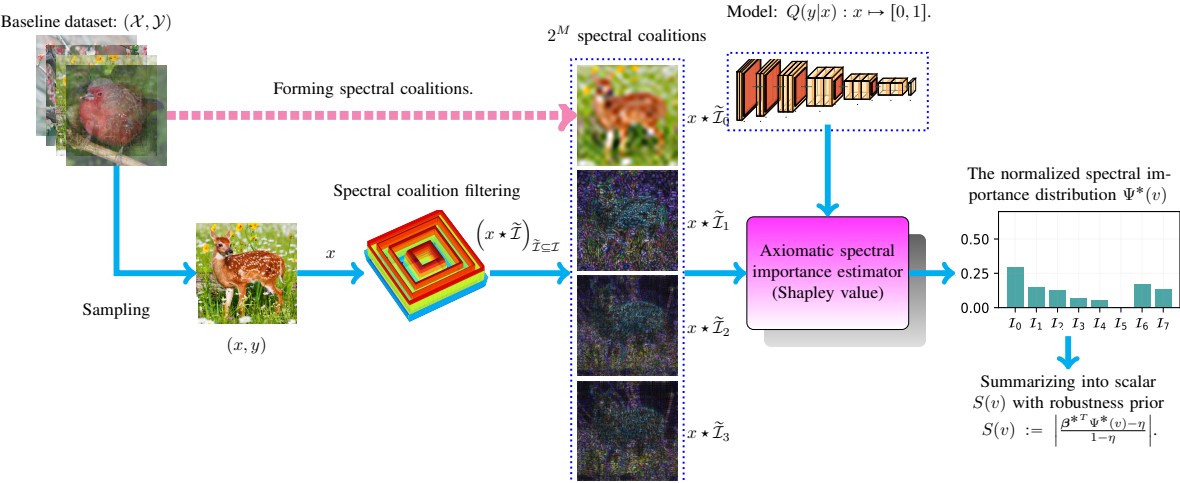

Figure 3: This diagram illustrates the overview of **I-ASIDE** with some baseline dataset $(\mathcal{X}, \mathcal{Y})$ (where $\mathcal{X}$ denotes images and $\mathcal{Y}$ denotes labels) and some classifier $Q(y|x) : x \mapsto [0, 1]$ which outputs a normalized probability distribution for $y \in \mathcal{Y}$. The images are sampled from the baseline dataset $(\mathcal{X}, \mathcal{Y})$ and fed into a spectral coalition filter to form $2^M$ spectral coalitions. The spectral coalitions are then used to compute the spectral importance distribution $\Psi^*(v)$. The normalized spectral importance distribution $\Psi^*(v)$ can be summarized into a robustness score $S(v)$.

- Providing an insight into understanding the adversarial vulnerability of models by examining spectral importance distributions (See the experiments in Figure 11);
- Studying model out-of-distribution (OOD) robustness when input domains shift (See the experiments in Figure 12). **I-ASIDE** can help to answer model robustness concerns from the perspective of feature representations in frequency-domain.

It is also notable that the potential applications are not restricted to the above fields. Other research fields such as data augmentation and self-supervised learning can also use **I-ASIDE** as a device to understand the dynamics of the learned feature representations of models. For example, in data augmentation research, **I-ASIDE** can interpret the training dynamics with ablation experiments by understanding how hyper parameters and augmentation tricks can affect the robustness of the learned features.

## 3  Related work

We summarize related works from four categories to provide research context in visual models: (1) Global interpretability, (2) model robustness, (3) frequency-domain research and (4) information theory in deep learning.

**Global interpretability**: Global interpretability summarizes the decision behaviours of models and provides a holistic view. In contrast, local interpretability merely provides explanations on the basis of instances (Sundararajan et al., 2017; Smilkov et al., 2017; Linardatos et al., 2020; Selvaraju et al., 2017; Arrieta et al., 2020; Zhou et al., 2016; Ribeiro et al., 2016; Lundberg & Lee, 2017; Lakkaraju et al., 2019; Guidotti et al., 2018; Bach et al., 2015; Montavon et al., 2019; Shrikumar et al., 2017). Due to the research scope, we do not unfold the literature review regarding local interpretability. Feature visualization with neuron maximization or class activation maximization can show the ideal inputs for specific neurons or classes by optimizing inputs, and help to understand the learned feature representations of models (Olah et al., 2017; Nguyen et al., 2019; Zeiler et al., 2010; Simonyan et al., 2013; Nguyen et al., 2016a;b). Yet, optimizing inputs by maximizing class scores or neuron activations can yield 'surrealistic' results which are not interpretable itself. Network dissection attempts to establish the connection between the functions of the units (e.g. a set of channels or a set of layers) in convolutional neural networks and some concepts – e.g. eyes or ears (Bau

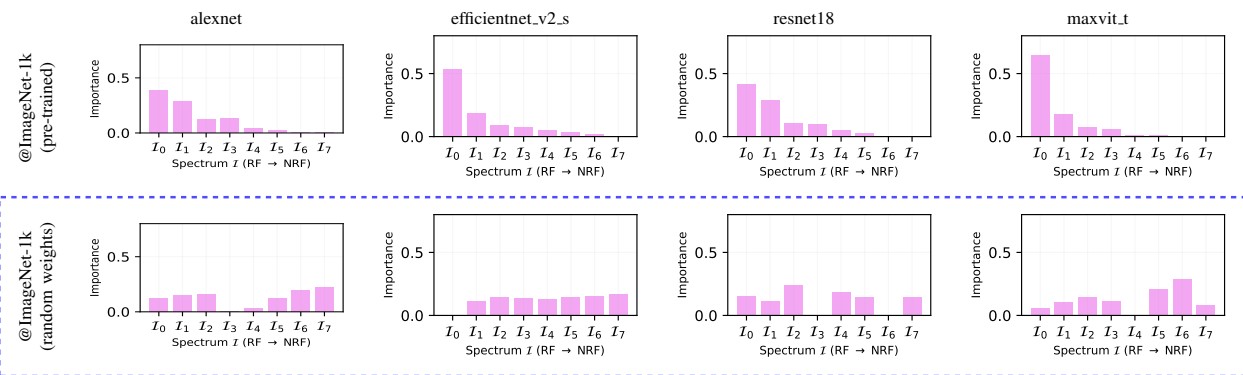

Figure 4: We showcase the spectral importance distributions from multiple models pre-trained on *ImageNet-1k*. We also include the models with random weights as a control marked by blue box. We have noticed that: (1) The spectral importance of the robustness of feature representations of trained models is non-uniform, and, (2) the spectral importance of the robustness of the feature representations of un-trained models is uniform.

et al., 2017). Concept-based approach measures the activations of networks with respect to concepts, and interpret networks at concept level (Kim et al., 2018; Ghorbani et al., 2019; Koh et al., 2020; Chen et al., 2020). Global input feature importance analysis summarizes the decisions of models by measuring how important the input features contribute to predictions (Altmann et al., 2010; Greenwell et al., 2018; Lundberg & Lee, 2017; Ribeiro et al., 2016; Simonyan et al., 2013; Sundararajan et al., 2017; Covert et al., 2020). For example, Covert et al. present SAGE by applying Shapley value theory (Shapley, 1997) to assign input features with importance for interpreting feature contributions. Yet, the implementations of these works are largely restricted to examining feature importance in the spatial domain. Interpreting the global feature importance in the spatial domain for images has inherited limits. For example, the showcase experiment of literature (Covert et al., 2020) provides the global pixel-wise explanation of a multi-layer perceptron (MLP) with the MNIST on spatial domain. The method assigns each pixel with an importance score and adds the scores over MNIST to form the global interpretation. However, the positions and angles of the objects in images can vary from image to image. Adding the pixel importance scores as global interpretation only offers very limited information. We only read that models pay attentions in the centers of the images in MNIST (Covert et al., 2020). Thus the interpretations in spatial domain often suffer from these limits and provide very limited insights regarding robustness. Our work differs from these prior works in that we interpret global model robustness in frequency domain. Frequency-domain transforms are not sensitive to the positions and angles of the objects of images. For example, the spectrum of rotated or flipped images remains identical. The frequency-invariant property in tandem with the correlation to perturbation robustness allows **I-ASIDE** to interpret model decision behaviors globally.

**Model robustness**: Szegedy et al. notice the robustness problem of deep models arising from adversarial perturbations. Thereafter, empirical observations demonstrate that boosting model robustness with adversarial training is at the cost of the performance degradation on model standard accuracy for visual tasks (Goodfellow et al., 2014). Later theoretical analyses show standard accuracy is at odds with model robustness (Zhang et al., 2019; Tsipras et al., 2018). Ilyas et al. argue features can be distinguished by their brittleness into robust features and non-robust features, and show that adversarial robustness relates to non-robust features. Our research is based on the above insights.

**Frequency-domain research**: Neural networks are non-linear parameterized signal processing filters. Investigating how neural networks respond to inputs in the frequency-domain can provide a unique insight into understanding its functions. Rahaman et al. approximate ReLU networks with piece-wise continuous linear functions in order to perform Fourier analysis. Their results suggest neural networks have a 'spectral bias' on smooth hypotheses (Raghu et al., 2017; Montufar et al., 2014; Rahaman et al., 2019). Xu et al. investigate the learning dynamics of neural networks in frequency-domain in their work 'F-Principle' (Xu et al., 2019a;b). Their work suggests that the learning behaviors of neural networks are spectral non-uniformity: Neural networks fit low-frequency components first, then high-frequency components later. Tsuzuku & Sato

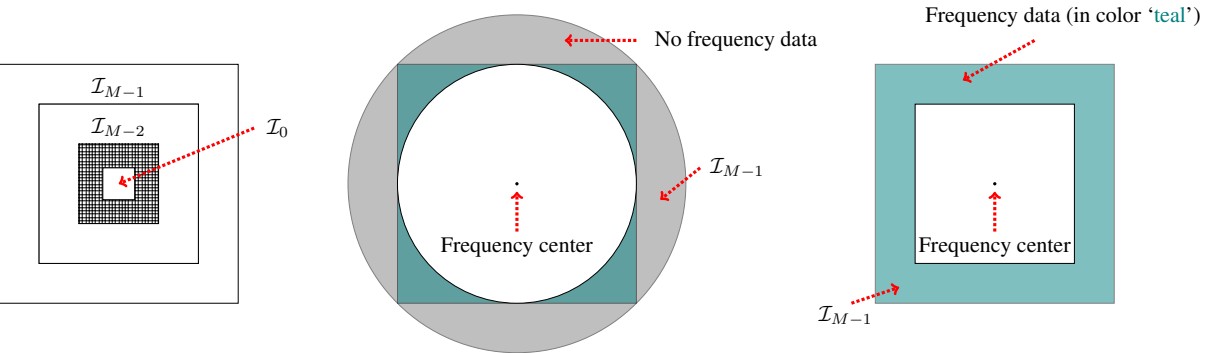

Figure 5: This shows the motivation we choose $\ell_\infty$ ball over $\ell_2$ ball in partitioning the frequency domain into the $M$ bands (i.e., $M$ 'spectral players') over 2D Fourier spectrum. The frequency data density of the spectral players with $\ell_\infty$ remains a constant. However, the frequency data density of the spectral players with $\ell_2$ is not a constant since some frequency components do not present. This motives us to empirically choose $\ell_\infty$ metric to form spectral players in implementation.

affirm convolutional neural networks have spectral non-uniformity regarding Fourier bases (Tsuzuku & Sato, 2019). Wang et al. conducts an empirical study of the connection between supervision signals and feature representations from the perspective of frequency (Wang et al., 2020). Kolek et al. propose 'CartoonX' and attempt to interpret the instance-wise decisions of image classifiers in wavelet domain (Kolek et al., 2022). Both **I-ASIDE** and CartoonX are the attempts to interpret the decision behaviors of models in frequency-domain. Our work differs from CartoonX in that: (1) We use Fourier transform instead of wavelet transform because the feature robustness to perturbations correlates with the Fourier spectrum of inputs, and (2) we assess the global decision behaviors over a population other than instance-wise.

**Information theory in deep learning**: In the Section 4, we use some theoretical results in information theory to justify and characterize the choice of the characteristic function in **I-ASIDE**. The optimization objectives of a variety of works from supervised learning and un-supervised learning can be unified under the view of maximizing variational mutual information (Poole et al., 2019; Nowozin et al., 2016; Kingma & Welling, 2013). Deep learning has achieved great success in enormous applications. Yet, the question regarding how models represent input features inside the black boxes remains mysterious. Tishby et al. answer this question by introducing information bottleneck theory for interpreting the learning dynamics of neural networks (Tishby et al., 2000; Saxe et al., 2019). The information bottleneck theory views models as an information encoding and decoding process. Models learn to encode (extract) the information from inputs relevant to labels while neglect irrelevant information (Saxe et al., 2019). Their work provides a motivation for **I-ASIDE**. The mutual information between the feature representations of models and labels reflect the abilities that models capture relevant information from input features. The decomposition of such mutual information with respect to frequency can reflect the spectral contributions to the learned information representations in black boxes. However, estimating the quantity of the mutual information is intractable since we do not have the knowledge regarding the joint distributions of feature representations and labels. Fortunately, Barber & Agakov show a variational lower bound between inputs and labels (Barber & Agakov, 2004). Qin et al. use the result and show that the training objective of the classifiers trained with usual cross-entropy is equivalent to estimating a variational bound of the mutual information between inputs and labels (Qin et al., 2021). We adopt their results in **I-ASIDE** to characterize the characteristic function design in Section 4.5.

## 4 Axiomatic spectral importance decomposition

We rigorously formulate the spectral importance decomposition problem as a fairness utility distribution problem in the context of coalition game. The fairness distribution (decomposition or division) is a problem

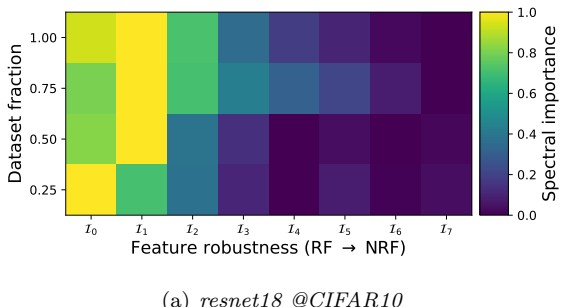

(a) *resnet18 @CIFAR10*

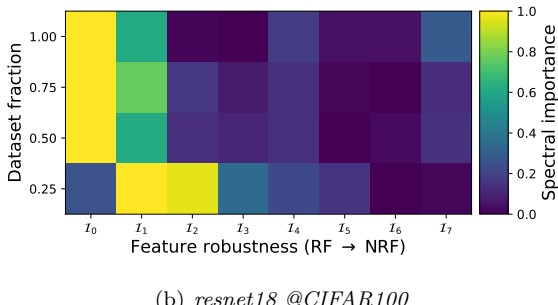

(b) *resnet18 @CIFAR100*

Figure 6: This demonstrates how the spectral importance distributions of feature representations change with respect to the training set sizes. We choose *resnet18* on *CIFAR10* and *CIFAR100*. The preliminary results show that there does not exist a unique pattern regarding how spectral importance distributions with respect to training set sizes.

in coalition game theory and refers to divide the payoffs among players in a coalitional game in which the division must satisfy a set of axioms (Aumann & Maschler, 1985; Yaari & Bar-Hillel, 1984; Aumann & Dombb, 2015; Hart, 1989; Roth, 1988). We directly deploy the Shapley value theory (Roth, 1988; Hart, 1989) to derive the solution to the decomposition problem in **I-ASIDE**.

We also theoretically justify and characterize the choice of the characteristic function in **I-ASIDE** by showing that **I-ASIDE** decomposes the mutual information between feature representations and labels for given classifiers with respect to spectrum. This theoretical characterization suggests **I-ASIDE** itself is interpretable at design level. Figure 3 shows the overview of **I-ASIDE**.

**Content organization**: We organize the content in the following order. In Section 4.1, we clarify notations and prepare preliminaries for theoretical analyses. In Section 4.2, we define a spectral coalition game to formulate the setting. In Section 4.3, we state the desirable axioms for fairness decomposition and state the result of Shapley value theory. In Section 4.4, we dive into the implementation of forming spectral coalitions in frequency-domain which is essential for deploying Shapley value theory in the spectral coalition game defined in Section 4.2. In Section 4.5, we define and theoretically characterize the design of the characteristic function of the defined spectral coalition game. The characterization suggests our method itself is interpretable at design level. In Section 4.6, we introduce an approach to summarize the measured spectral importance distributions into scalar scores. In Section 4.7, we theoretically analyze the error bound of our method and show the experiments to prove the analysis.

### 4.1 Notations and preliminaries

We adopt the notation conventions from topology and functional analysis (Yosida, 2012; O'Searcoid, 2006; Simmons, 1963). We also follow the conventions in related literature of deep learning community such as variational methods in deep learning. We clarify the notations below.

Let $(\mathcal{X}, \mathcal{Y})$ be some image dataset where $\mathcal{X}$ denotes images and $\mathcal{Y}$ denotes discrete labels. Let $S_{y'}(x; \theta) : x \mapsto \mathbb{R}$ be the logit output of some parameterized classifier $S$ with some input $x$ with respect to class $y' \in \mathcal{Y}$ where $\theta$ is parameters. We use $Q$ and $P$ to differ 'prediction' and 'ground-truth'. Let $Q(y|x; \theta) : (x, y) \mapsto [0, 1]$ be the conditional probability expression of the classifier $S$:

$$Q(y|x; \theta) \stackrel{def}{=} \frac{e^{S_y(x;\theta)}}{\sum\limits_{y' \in \mathcal{Y}} e^{S_{y'}(x;\theta)}} \tag{1}$$

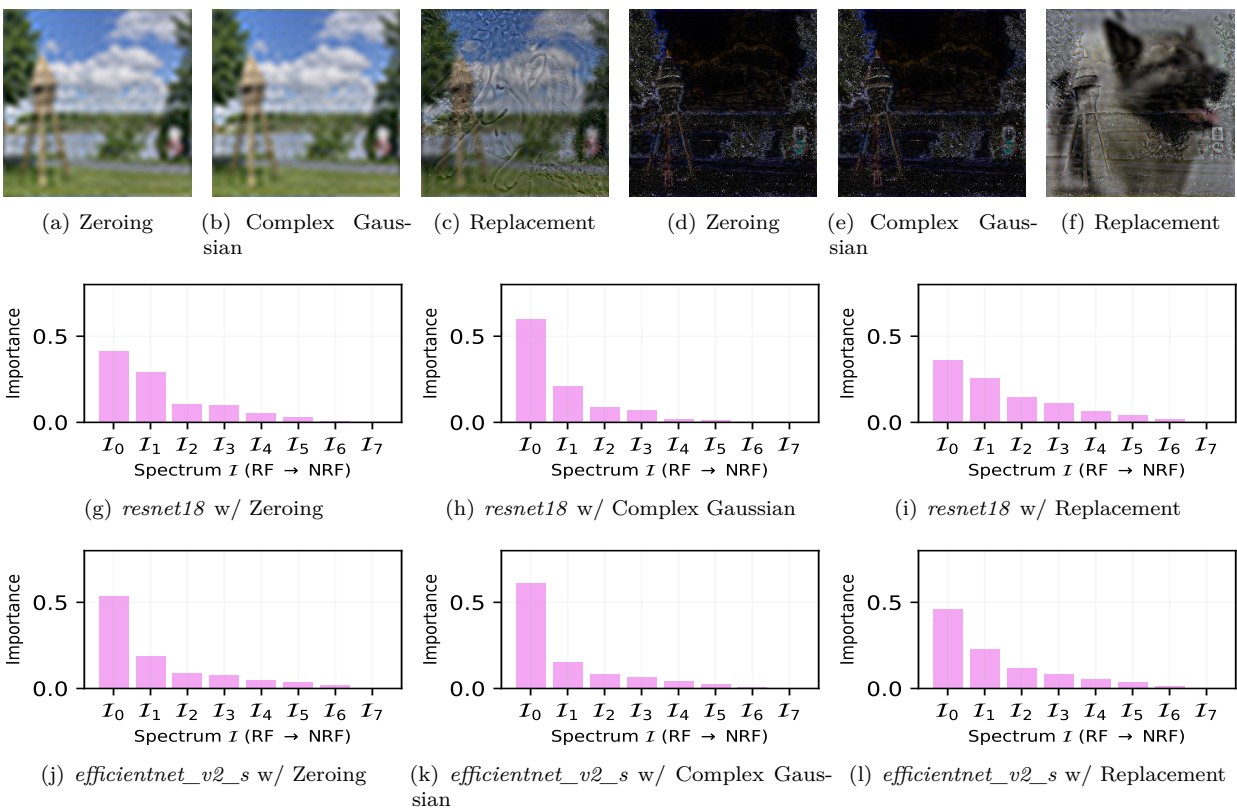

Figure 7: This shows the filtering examples with three masking (absence assignment) strategies: (1) Assigning the spectral absences with constant zeros (Zeroing), (2) assigning the spevtral absences with Gaussian noise (Complex Gaussian) and (3) randomly sampling spectral components from the same image datasets (Replacement). The standard complex Gaussian distribution is given by: $\mathcal{N}(0, \frac{1}{2}) + i\mathcal{N}(0, \frac{1}{2})$. The figures (a), (b) and (c) show the coalition filtering results with the spectral coalition: $\{\mathcal{I}_0\}$. The figures (d), (e) and (f) show the coalition filtering results with the spectral coalition: $\{\mathcal{I}_1, \mathcal{I}_2, \mathcal{I}_3, \mathcal{I}_4, \mathcal{I}_5, \mathcal{I}_6, \mathcal{I}_7\}$. The figures (g) to (l) show the examples of the measured spectral importance distributions of a *resnet18* and a *efficientnet_v2_s* (both are pre-trained on *ImageNet*) with the three assignment strategies.

where $\theta$ is the parameters. Clearly, $Q(y|x; \theta) \in [0, 1]$. Let $Q(\mathcal{Y}|x; \theta) : x \mapsto [0, 1]^{|\mathcal{Y}|}$ be the predicted distribution for some given image $x \in \mathcal{X}$ over $\mathcal{Y}$:

$$Q(\mathcal{Y}|x; \theta) \stackrel{def}{=} \left( Q(y|x; \theta) \right)_{y \in \mathcal{Y}} \in [0, 1]^{|\mathcal{Y}|} \tag{2}$$

where $|\mathcal{Y}|$ is the cardinality (the number of elements) of set $\mathcal{Y}$. We use $P(y'|x)$ to denote the prior probability of image $x$ for given class $y'$. In terms of one-hot labels, suppose the ground-truth label of image $x$ is $y$, the distribution $P(y'|x)$ is given by:

$$P(y'|x) = \begin{cases} 1, & \text{if } y' = y, \\ 0, & \text{if } y' \neq y \end{cases}. \tag{3}$$

Accordingly, $P(\mathcal{Y}|x) = (0, \cdots, 0, 1, 0, \cdots, 0)$. We use the convention from topology (analysis) to denote the collection of the distributions $Q(\mathcal{Y}|x; \theta)$ over a set $\mathcal{X}$ as:

$$Q(\mathcal{X}; \theta) \stackrel{def}{=} \{Q(\mathcal{Y}|x; \theta)|x \in \mathcal{X}\} \tag{4}$$

where $\theta$ denotes the parameter of classifier $Q$. Since the distributions of $Q$ over the dataset $\mathcal{X}$ represent the representations of the data in the dataset, we also refer $Q(\mathcal{X}; \theta)$ as the representations of set $\mathcal{X}$ for given

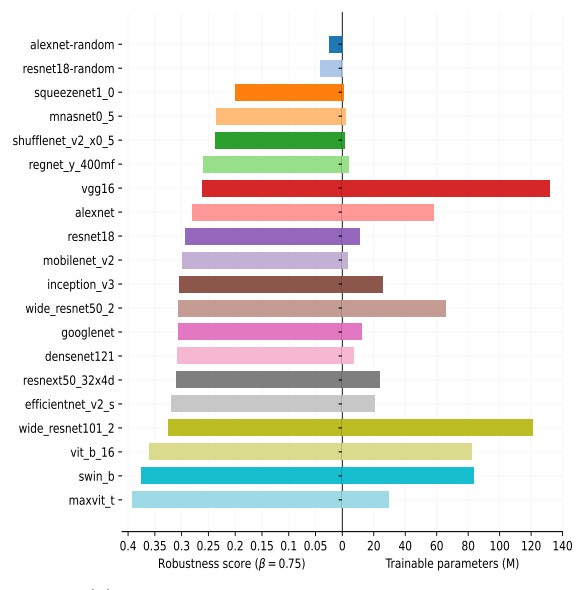
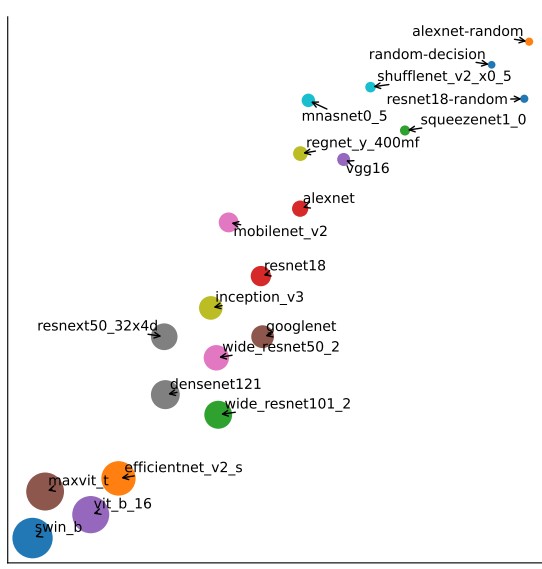

(a) Spectral importance score comparison

(b) Spectral importance distribution projection

Figure 8: This is an application showcase (A1) in research with two experiments. The left bidirectional chart showcases to model spectral importance scores with respect to trainable parameters. The right figure showcases to visualize model robustness by projecting the spectral importance distributions with t-SNE. The shape sizes correspond to robustness (the larger is the better). The experiments are performed on dataset *ImageNet*. The results also correlate with the experiments in Figure 11 regarding adversarial perturbations. In our experiments, the numbers of the trainable parameters of models do not play a crucial role on model robustness.

classifier $Q$. We ignore the parameter $\theta$ in all notations to maintain depiction succinct. For example, we denote $Q(y|x;\theta)$ as $Q(y|x)$, $Q(\mathcal{Y}|x;\theta)$ as $Q(\mathcal{Y}|x)$ and $Q(\mathcal{X};\theta)$ as $Q(\mathcal{X})$.

## 4.2 Spectral coalition game

The probability predictions of an image classifier can be viewed as the worth (utility) scores in a coalition game with spectral players. The spectral players are the radial spectral regions in such a spectral coalition game (See Figure 1). The coalition game is defined by some image classifier. Spectral players *cooperate*, *interact*, and *contribute* to the decisions of the classifiers with various bargaining powers. The worth division should satisfy a set of axioms: *efficiency*, *symmetry*, *linearity* and *dummy player* (Roth, 1988; Hart, 1989; Winter, 2002). Shapley value theory is the unique solution to such a division.

We now introduce 'spectral coalition game'. Let $Q(y|x) : x \mapsto [0,1]$ be some image classifier which predicts the discrete category probability distribution of some $x$ for $y \in \mathcal{Y}$. Let $[0,1]$ be the normalized 1-dimensional radial spectrum (See Figure 1 and Figure 5). The radial spectrum is partitioned into $M$ equispaced regions (See Figure 5). Each region is a $\ell_\infty$ ball with various radius. Each partition then is a '*spectral player*'. The $i$-th spectral player is denoted as $\mathcal{I}_i$ (where $i \in [M] \stackrel{\text{def}}{=} \{0,1,\cdots,M-1\}$). The $M$ spectral players constitute a player set $\mathcal{I} := \{\mathcal{I}_i\}_{i=0}^{M-1}$. The consideration of using $\ell_\infty$ ball is because: (1) It is easier to handle the distant pixels at implementation and (2) the pixel density in the region remains constant compared with $\ell_p$ ($1 < p < \infty$). Please refer to Figure 5. Let $v : 2^{\mathcal{I}} \mapsto \mathbb{R}$ be some characteristic function which sends $2^{\mathcal{I}}$ to the field $\mathbb{R}$ such that $v(\varnothing) = 0$. The map $v$ is given as the form of the statistical expectation of some $Q(y|x)$ over some baseline dataset $(\mathcal{X},\mathcal{Y})$. We will show that this choice can be justified in that the characteristic function $v$ measures a variational lower bound (Poole et al., 2019; Barber & Agakov, 2004) of the mutual information between images and labels in Section 4.5. We define a coalition game $(\mathcal{I},v)$. Let $\psi_i(\mathcal{I},v)$ be the spectral importance component for $\mathcal{I}_i$. Let $\Psi(v) := (\psi_i)_{i \in [M]}$ be the collection of $\psi_i(\mathcal{I},v)$.

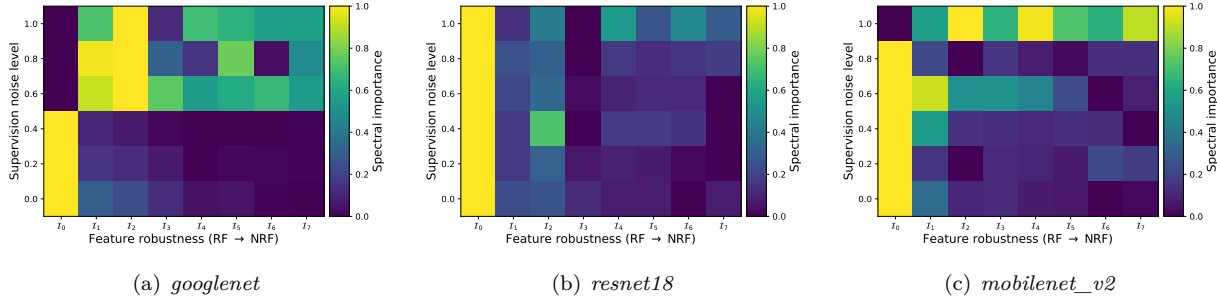

(a) *googlenet*  (b) *resnet18*  (c) *mobilenet_v2*

Figure 9: This is an application showcase (A2) demonstrating how models respond to the label noise of a human-annotated dataset in training.

## 4.3 Axioms and decomposition

Dividing the 'payoffs' amid players in coalition games can exist multiple non-unique schemes. A fair division scheme is said that the scheme satisfies a set of desirable fairness axioms (Moulin, 1992; Roth, 1988; Hart, 1989; Winter, 2002; Shapley & Shubik, 1969; van den Brink, 2002). Shapley value theory is a solution concept which uniquely satisfies a set of desirable fairness axioms. We formulate the spectral importance decomposition as a fairness division problem in 'spectral coalition game'. We state the essential axioms below.

***Symmetry* axiom**: Let $\widetilde{\mathcal{I}} \in 2^{\mathcal{I}}$ be some spectral player coalition. For $\forall\ \mathcal{I}_i, \mathcal{I}_j \in \mathcal{I} \wedge \mathcal{I}_i, \mathcal{I}_j \notin \widetilde{\mathcal{I}}$, the statement $v(\widetilde{\mathcal{I}} \cup \{\mathcal{I}_i\}) = v(\widetilde{\mathcal{I}} \cup \{\mathcal{I}_j\})$ implies $\psi_i(\mathcal{I}, v) = \psi_j(\mathcal{I}, v)$. This axiom relabels the statement '*equal treatment of equals*' principle mathematically. This axiom states that the 'names' of players should have no effect on the 'treatments' by the characteristic function in coalition games (Roth, 1988).

***Linearity* axiom**: Let $u$ and $v$ be two characteristic functions. Let $(\mathcal{I}, u)$ and $(\mathcal{I}, v)$ be two coalition games. Let $(u + v)(\widetilde{\mathcal{I}}) := u(\widetilde{\mathcal{I}}) + v(\widetilde{\mathcal{I}})$ where $\widetilde{\mathcal{I}} \in 2^{\mathcal{I}}$. The divisions of the new coalition game $(\mathcal{I}, u + v)$ should satisfy: $\psi_i(\mathcal{I}, u + v) = \psi_i(\mathcal{I}, u) + \psi_i(\mathcal{I}, v)$. This axiom is also known as '*additivity* axiom' and guarantees the uniqueness of the solution of dividing payoffs amid players (Roth, 1988).

***Efficiency* axiom**: This axiom states that the sum of the divisions of all players must be summed to the worth of player set (the grand coalition): $\sum_{i=0}^{M-1} \psi_i(\mathcal{I}, v) = v(\mathcal{I})$.

***Dummy player* axiom**: A dummy player (null player) $\mathcal{I}_*$ is the player who has no contribution such that: $\psi_*(\mathcal{I}, v) = 0$ and $v(\widetilde{\mathcal{I}} \cup \{\mathcal{I}_*\}) \equiv v(\widetilde{\mathcal{I}})$ for $\forall\ \mathcal{I}_* \notin \widetilde{\mathcal{I}} \wedge \mathcal{I}_* \subseteq \mathcal{I}$.

In the Shapley value theory literature (Roth, 1988), the *efficiency* axiom and the *dummy player* axiom are also relabeled as *carrier* axiom. We directly deploy Shapley value theory to decompose the predictions of models amongst spectral players. The decompositions are uniquely given by:

$$\psi_i(\mathcal{I}, v) = \sum_{\widetilde{\mathcal{I}} \subseteq \mathcal{I} \backslash \mathcal{I}_i} \frac{1}{M} \binom{M-1}{|\widetilde{\mathcal{I}}|}^{-1} \left\{ v(\widetilde{\mathcal{I}} \cup \{\mathcal{I}_i\}) - v(\widetilde{\mathcal{I}}) \right\} \tag{5}$$

where $\frac{1}{M}\binom{M-1}{|\widetilde{\mathcal{I}}|}^{-1}$ gives the probability of sampling a spectral player $\mathcal{I}_i$ and a spectral player coalition $\widetilde{\mathcal{I}}$.

## 4.4 Spectral coalition filtering

To implement the spectral coalitions in the spectral coalition game defined in Section 4.2, we must implement spectral coalition filtering. We implement the spectral coalition with ideal radial multi-band-pass digital signal filtering (Oppenheim, 1978; Roberts & Mullis, 1987; Pei & Tseng, 1998). The multi-band-pass digital signal filtering refers that the signal components in pass-bands are preserved while the signal components in stop-bands are suppressed. We thus can use the 'pass-bands' and 'stop-bands' to represent the 'presences' and 'absences' of the players in coalitions (See Figure 5). We state the implementation below.

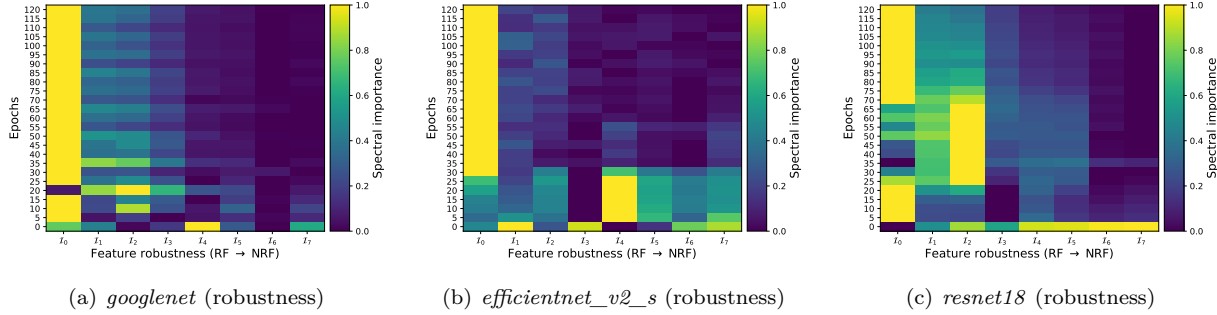

(a) *googlenet* (robustness)  (b) *efficientnet_v2_s* (robustness)  (c) *resnet18* (robustness)

Figure 10: This is an application showcase (A3) in research demonstrating how the robustness of feature representations of models evolves with respect to training epochs. We also provide the loss curves with respect to epochs in Appendix A.5.

Let $\mathscr{F}$ be the Discrete Fourier transform (DFT) operator and $\mathscr{F}^{-1}$ be inverse DFT (IDFT) operator (Tan & Jiang, 2018). The DFT operator $\mathscr{F}$ and IDFT operator $\mathscr{F}^{-1}$ are defined in Appendix A.2. We take some spectral coalition $\widetilde{\mathcal{I}} \in 2^{\mathcal{I}}$ and crop the frequency components not present in $\widetilde{\mathcal{I}}$ by using a channel-wise plane multi-band-pass filter on frequency-domain (Oppenheim, 1978; Roberts & Mullis, 1987; Castleman, 1996; Jähne, 2005). The multi-band-pass mask matrices of allowing pass-bands and suppressing stop-bands are referred as 'transfer functions' or 'filters' in the literature of digital signal processing. Let $\widetilde{\mathcal{I}} \in 2^{\mathcal{I}}$ be some spectral player coalition in some spectral coalition game $(\mathcal{I}, v)$. Let $\mathbb{T}(\widetilde{\mathcal{I}}) = \big(\mathbb{T}(\widetilde{\mathcal{I}})(m,n)\big)_{(m,n)\in[M]\times[N]} \in \mathbb{R}^{M\times N}$ be the filter transfer function for the spectral coalition $\widetilde{\mathcal{I}}$ such that:

$$\mathbb{T}(\widetilde{\mathcal{I}})(m,n) = \begin{cases} 1, & \text{if the frequency point (m,n) in } \widetilde{\mathcal{I}}, \\ 0, & \text{otherwise} \end{cases} \tag{6}$$

where $M \times N$ is the dimension of the image. We define a binary operator '$\star$' over the field $\mathbb{R}$ to represent the coalitional filtering by:

$$x \star \widetilde{\mathcal{I}} \stackrel{def}{=} \mathscr{F}^{-1}\left[\underbrace{\mathscr{F}(x) \odot \mathbb{T}(\widetilde{\mathcal{I}})}_{\text{Spectral presence}} + \underbrace{\boldsymbol{b} \odot (\mathbb{1} - \mathbb{T}(\widetilde{\mathcal{I}}))}_{\text{Spectral absence}}\right] \tag{7}$$

where '$\odot$' denotes Hadamard product (Horn, 1990; Horadam, 2012), $\mathbb{1} \in \mathbb{R}^{M\times N}$ denotes an all-ones matrix and $\boldsymbol{b} \in \mathbb{C}^{M\times N}$ represents the assignments of the absences of spectral players over complex field. At our implementation, we empirically set $\boldsymbol{b} = \boldsymbol{0}$.

**Absence baseline**: The term 'baseline' in attribution analysis context refers to the absence assignments of players (Sundararajan et al., 2017; Shrikumar et al., 2017; Binder et al., 2016). For example, if we use 'zeros' to represent the absence of players, the 'zeros' are dubbed as 'baseline'. Intuitively, 'baseline' is the 'pretext' for being 'absent' in the coalition game (Sundararajan et al., 2017).

**Spectral player absence assignment**: In the implementation of the coalition filtering, the masking (assignment) strategies of the absences of the spectral players in coalition filtering can change the distributions of filtered images. There exist multiple choices for the assignments of the absences of spectral layers in coalition filtering design: (1) Assigning to constant zeros (Zeroing), (2) assigning to complex Gaussian noise (Complex Gaussian) and (3) assigning to the corresponding frequency components randomly sampled from other images at the same dataset (Replacement). When we adopt 'Complex Gaussian' strategy, the $\boldsymbol{b}$ equation (7) is sampled from some *i.i.d.* complex Gaussian distribution: $\mathcal{N}(\mu, \frac{\sigma^2}{2}) + i\mathcal{N}(\mu, \frac{\sigma^2}{2})$. When we adopt the 'Replacement' strategy, the equation (7) changes to: $\boldsymbol{b} = \mathscr{F}(x^*)$ (where $x^* \sim \mathcal{X}$ is a randomly sampled image from some set $\mathcal{X}$). In our implementation, we simply choose 'zeroing' as our filtering strategy: $\boldsymbol{b} = \boldsymbol{0}$. In Figure 7, we show the filtered image examples by using the above three strategies and also show the examples

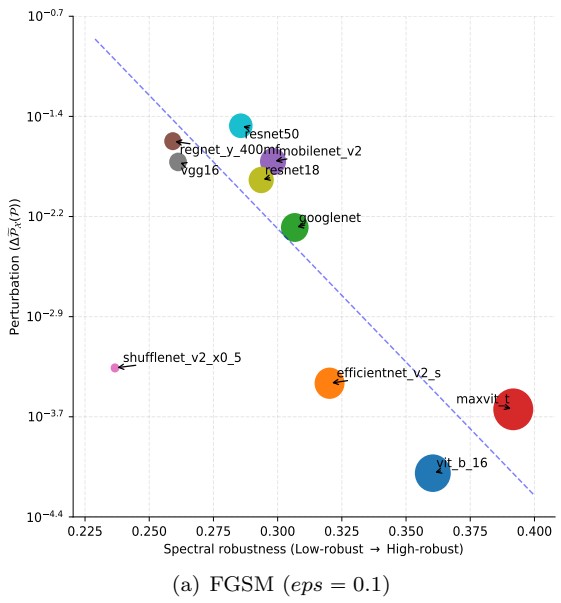
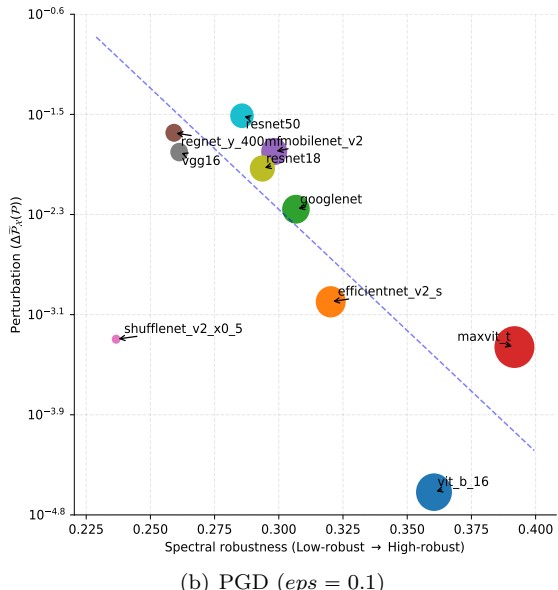

(a) FGSM ($eps = 0.1$)                    (b) PGD ($eps = 0.1$)

Figure 11: This is an application showcase (A4) in research demonstrating that there is a correlation between adversarial perturbations and the spectral importance scores using **I-ASIDE**. The adversarial perturbations are measured by the statistical expectations of prediction probability variations (See Section 5.4). We choose multiple pretrained models (on *ImageNet*) and perform un-targeted FGSM/PGD attacking with $eps = 0.1$. The adversarial samples carry a high proportion of non-robust features (Ilyas et al., 2019) (See Figure 2). For most models, adversarial perturbations are negatively proportional to spectral importance scores. The circle sizes are proportional to spectral importance scores. It is also notable there are some outliers (e.g. *shufflenet*), and this implies the perturbation-based robustness research can merely capture an aspect of the holistic robustness of models. The full results are provided in Appendix A.6.

of measured spectral importance distributions. In our empirical observations, the three strategies have rather similar performance. In this research, we do not unfold the discussions regarding the masking strategy choices. We will investigate the masking strategy choices in our future work.

### 4.5 Characteristic function design and characterization

Let $Q(y|x) : x \mapsto [0, 1]$ be some image classifier. For a sample $x$, $Q(y|x)$ sends the $x$ to a discrete category probability distribution for $y \in \mathcal{Y}$. Let $Q(\mathcal{X})$ denote the representations of image set $\mathcal{X}$ with given classifier $Q$. We design the characteristic function on the classifier $Q$ over dataset $\mathcal{D} := (\mathcal{X}, \mathcal{Y})$ as:

$$v(\widetilde{\mathcal{I}}; Q, \mathcal{D}) := \mathop{\mathbb{E}}_{x,y \sim (\mathcal{X}, \mathcal{Y})} \left\{ \log Q(y|x \star \widetilde{\mathcal{I}}) - \log Q(y|x \star \varnothing) \right\} \tag{8}$$

where $\forall \widetilde{\mathcal{I}} \in 2^{\mathcal{I}}$. The characteristic function $v$ is a function of spectral coalitions ($\widetilde{\mathcal{I}} \in 2^{\mathcal{I}}$) over given dataset $(\mathcal{X}, \mathcal{Y})$ and classifier $Q$. The map $v$ also satisfies $v(\varnothing) = 0$ if the absences of spectral players are assigned to zeros in coalition filtering (See Section 4.4). We denote $v(\widetilde{\mathcal{I}}; Q, \mathcal{D})$ as $v(\widetilde{\mathcal{I}})$ to maintain depiction succinct.

We justify the choice of $v$ by showing two claims:

Claim 1 An image classifier $Q$ trained with usual cross-entropy loss estimates a variational lower bound of the mutual information between images $\mathcal{X}$ and labels $\mathcal{Y}$ (Qin et al., 2021; Barber & Agakov, 2004);

Claim 2 The map $v$ defined on the top of the $Q$ and $(\mathcal{X}, \mathcal{Y})$ measures the mutual information between the representations $Q(\mathcal{X})$ and labels $\mathcal{Y}$ which also is a variational bound of the mutual information between images $\mathcal{X}$ and labels $\mathcal{Y}$.

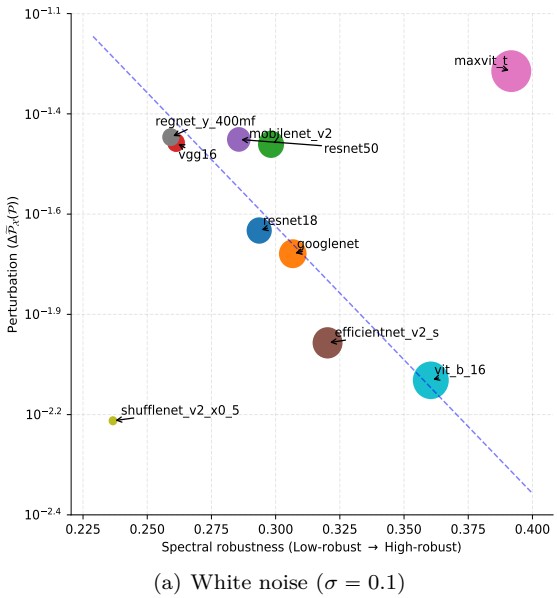 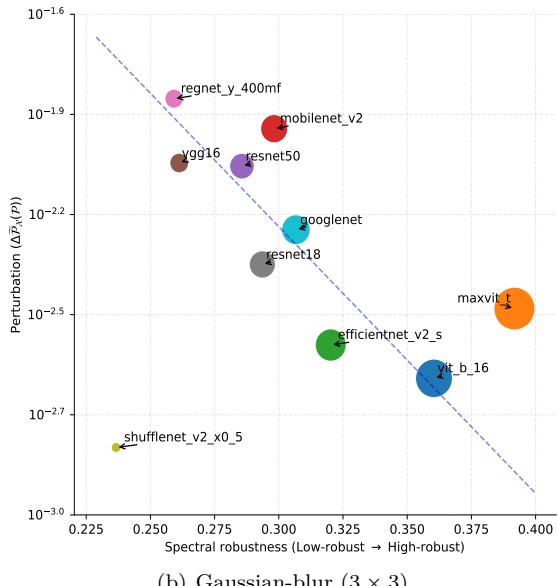

(a) White noise ($\sigma = 0.1$)      (b) Gaussian-blur ($3 \times 3$)

Figure 12: This is an application showcase (A5) in research demonstrating that there is a correlation between out-of-distribution (OOD) perturbations and the spectral importance scores using **I-ASIDE**. The OOD perturbations are measured by the statistical expectations of prediction probability variations (See Section 5.5). We choose multiple pretrained models (on *ImageNet*) and evaluate the perturbations with two OOD transforms: Gaussian ($\sigma = 0.1$) and Gaussian-blur ($kernel = 3 \times 3$). The results show that the OOD perturbations are negatively proportional to spectral importance scores. The circle sizes are proportional to spectral importance scores. It is also notable there are some outliers (e.g. *maxvit_t*), and this implies the perturbation-based robustness research can merely capture an aspect of the holistic robustness of models.

If the above two claims hold, the map $v$ over a classifier $Q$ can assess the feature representation quality of the classifier. We first show that the 'Claim 1' holds. Let $(\mathcal{X}, \mathcal{Y})$ be some dataset. Let $Q^*(y'|x) : (x, y') \mapsto [0, 1]$ be an image classifier which sends an image $x \in \mathcal{X}$ to a probability for given class $y' \in \mathcal{Y}$. Let $P(y'|x)$ be the prior probability of image $x \in \mathcal{X}$ for given class $y' \in \mathcal{Y}$. In terms of one-hot labels, suppose the ground-truth label of image $x$ is $y$, the distribution $P(y'|x)$ is given by:

$$P(y'|x) = \begin{cases} 1, & \text{if } y' = y, \\ 0, & \text{if } y' \neq y \end{cases}.$$

(9)

Let $\mathcal{L}_{CE}(Q^*, x, y)$ be the cross-entropy loss over the dataset:

$$\mathcal{L}_{CE}(Q^*, x, y) \stackrel{def}{=} \mathop{\mathbb{E}}_{x, y \sim (\mathcal{X}, \mathcal{Y})} \left[ \sum_{y' \in \mathcal{Y}} -P(y'|x) \log Q^*(y'|x) \right].$$

(10)

The optimization objective of some image classifier with usual cross-entropy loss $\mathcal{L}_{CE}(Q^*, x, y)$ in learning is:

$$\inf_{Q^*} \mathcal{L}_{CE}(Q^*, x, y) = \inf_{\forall \ Q^*} \mathop{\mathbb{E}}_{x, y \sim (\mathcal{X}, \mathcal{Y})} \left[ \sum_{y' \in \mathcal{Y}} -P(y'|x) \log Q^*(y'|x) \right]$$

(11)

$$= \inf_{\forall \ Q^*} \mathop{\mathbb{E}}_{x, y \sim (\mathcal{X}, \mathcal{Y})} -\log Q^*(y|x)$$

(12)

$$= \sup_{\forall \ Q^*} \mathop{\mathbb{E}}_{x, y \sim (\mathcal{X}, \mathcal{Y})} \log Q^*(y|x)$$

(13)

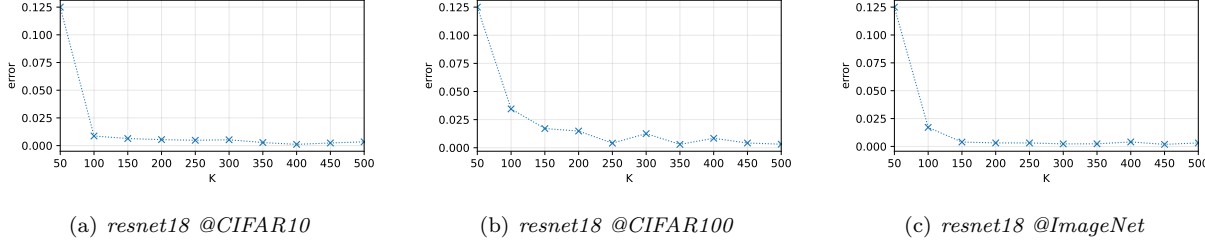

(a) *resnet18 @CIFAR10*      (b) *resnet18 @CIFAR100*      (c) *resnet18 @ImageNet*

Figure 13: This shows how the relative estimation errors converge with respect to the numbers of samples $K$. The errors are measured by: $\frac{1}{M}||\Psi^{(i+1)}(v) - \Psi^{(i)}(v)||_1$ where $\Psi^{(i)}(v)$ denotes the $i$-th measured spectral importance distribution with respect to characteristic function $v$. The experiments are conducted on *CIFAR10*, *CIFAR100* and *ImageNet* with *resnet18*.

**Lemma 4.1.** *B.A. Feature Representation Variational Mutual Information Bound. Let $(\mathcal{X}, \mathcal{Y})$ be some dataset. Let $Q^*(y|x)$ be some classifier. Let $Q$ be the optimal model optimized on the dataset with cross-entropy loss. Let $Q(\mathcal{X})$ be the representations of the set $\mathcal{X}$. Let $H(\mathcal{Y})$ be the differential entropy of $\mathcal{Y}$. Let $\mathbb{I}(\mathcal{X}; \mathcal{Y})$ be the mutual information between images and labels. The below variational bound holds and we refer the LHS as 'B.A. feature representation variational mutual information bound' or 'B.A. mutual information bound'. The bound is:*

$$\underbrace{\mathbb{I}_{BA}(Q(\mathcal{X}); \mathcal{Y}) \overset{def}{=} \sup_{Q^*} \mathop{\mathbb{E}}_{x,y \sim (\mathcal{X}, \mathcal{Y})} \log Q^*(y|x) + H(\mathcal{Y})}_{\textit{B.A. mutual information bound}} \leqslant \mathbb{I}(\mathcal{X}; \mathcal{Y}) \tag{14}$$

*where the notation $\mathbb{I}_{BA}(Q(\mathcal{X}); \mathcal{Y})$ is read as: The B.A. variational mutual information bound of dataset $(\mathcal{X}, \mathcal{Y})$ on classifier $Q$. The proof is provided in Appendix A.1.*

Clearly, the Lemma 4.1 shows 'Claim 1' holds:

$$\inf_{Q^*} \mathcal{L}_{CE}(Q^*, x, y) = \mathbb{I}_{BA}(Q(\mathcal{X}); \mathcal{Y}) - H(\mathcal{Y}). \tag{15}$$

We now can show 'Claim 2' holds. Set $\mathcal{X} \star \widetilde{\mathcal{I}} := \{x \star \widetilde{\mathcal{I}} | x \in \mathcal{X}\}$. Applying the Lemma 4.1 into the equation (8), we can rewrite the characteristic function $v$ as:

$$v(\widetilde{\mathcal{I}}) := \underbrace{\mathbb{I}_{BA}(Q(\mathcal{X} \star \widetilde{\mathcal{I}}), \mathcal{Y})}_{\text{B.A.bound}} - \underbrace{\mathbb{I}_{BA}(Q(\mathcal{X} \star \varnothing), \mathcal{Y})}_{\text{constant}} \tag{16}$$

which clearly measures the mutual information between between representations and labels, and also the B.A. variational bound of the mutual information between images and labels. The two claims give a theoretical interpretation and justify the design of the characteristic function in **I-ASIDE**. This implies **I-ASIDE** itself is interpretable at design level.

### 4.6 Summarizing spectral importance distribution

We next discuss how to summarize the distributions into real scalars. Let $S(v) : \Psi(v) \mapsto [0, 1]$ be a functional on field $\mathbb{R}$ which summarizes the derived spectral importance distributions into real scalars. There are several choices to summarize according to our needs. We are interested in feature representation robustness. We summarize the scores by summing with a feature representation robustness prior series as weights: The learned feature representations dominated by low-frequency components are more robust than those feature representations dominated by high-frequency components.

Intuitively, we can rank the importance of the $M$ spectral players as: $\widetilde{\boldsymbol{\beta}} := (0, -1, -2, \cdots, -(M-1))$. We use an exponential transform with base $\beta$ to convert such a ranking vector $\widetilde{\boldsymbol{\beta}}$ into a non-negative series:

$\boldsymbol{\beta} = (\beta^0, \beta^1, \cdots, \beta^{M-1})$ where $0 < \beta < 1$. The weighting vector $\boldsymbol{\beta}$ preserves the ranking information but the weights vary according to the choice of $\beta$. Set $\Psi^*(v) = \frac{\Psi(v) - \min \Psi(v)}{||\Psi(v) - \min \Psi(v)||_1}$ for normalizing $\Psi(v)$. The normalized $\Psi^*(v)$ can be summarized by the inner product with the prior weighting vector $\boldsymbol{\beta}$: $\boldsymbol{\beta}^T \Psi^*(v)$. We further take into account the random decision as a baseline by the same model with randomly initialized weights:

$$\left| \boldsymbol{\beta}^T \Psi^*(v) - \underbrace{\mathbb{E}_{\widetilde{v}}[\boldsymbol{\beta}^T \Psi^*(\widetilde{v})]}_{\text{baseline}} \right| \tag{17}$$

where $\widetilde{v}$ denotes the characteristic function of some model with randomly initialized weights and $\mathbb{E}_{\widetilde{v}}[\boldsymbol{\beta}^T \Psi^*(\widetilde{v})]$ denotes the statistical expectation of the measured distributions when the model over all random weights. We set:

$$\mathbb{E}_{\widetilde{v}}[\boldsymbol{\beta}^T \Psi^*(\widetilde{v})] \approx \boldsymbol{\beta}^T \frac{\mathbb{1}}{M}. \tag{18}$$

The approximation equation (18) holds true because the motivation experiments in Figure 4 show that the measured spectral importance distributions of models with randomized weights exhibit uniformity. The result can further be normalized into $[0, 1]$ by:

$$S(v) := \frac{\left| \boldsymbol{\beta}^T \Psi^*(v) - \boldsymbol{\beta}^T \frac{\mathbb{1}}{M} \right|}{\sup \left| \boldsymbol{\beta}^T \Psi^*(v) - \boldsymbol{\beta}^T \frac{\mathbb{1}}{M} \right|} = \left| \frac{\boldsymbol{\beta}^{*T} \Psi^*(v) - \eta}{1 - \eta} \right| \tag{19}$$

where $\beta \in (0, 1)$, $\boldsymbol{\beta}^* = \frac{\boldsymbol{\beta}}{||\boldsymbol{\beta}||_2}$ and $\eta = \frac{1}{M} \frac{||\boldsymbol{\beta}||_1}{||\boldsymbol{\beta}||_2}$. The deduction of this formula is provided in Appendix A.3.

### 4.7 Estimation of error upper bound

Suffering from the limit of computational resources, the statistical expectation is evaluated by using Monte Carlo sampling. We analyze the error bound by taking $K$ samples. Set $\Delta v(\tilde{\mathcal{I}}, \mathcal{I}_i) := v(\tilde{\mathcal{I}} \cup \{\mathcal{I}_i\}) - v(\tilde{\mathcal{I}})$. Let $\psi_i^*$ and $\Delta v^*(\tilde{\mathcal{I}}, \mathcal{I}_i)$ be estimations with $K$ samples using Monte Carlo sampling. The error bound with $\ell_1$ norm is given by:

$$\epsilon_K \stackrel{\text{def}}{=} \sup_i \mathbb{E}_{(\mathcal{X}, \mathcal{Y})} ||\psi_i^*(\mathcal{I}, v) - \psi_i(\mathcal{I}, v)||_1 \leqslant 2^{M-1} \cdot \left\{ \frac{Var(\Delta v^*)}{K} \right\}^{\frac{1}{2}} \tag{20}$$

where $Var(\Delta v^*)$ gives the upper bound of the variance of $\Delta v^*(\tilde{\mathcal{I}}, \mathcal{I}_i)$. Clearly, $\lim \epsilon_K \mapsto 0$ as $K \mapsto \infty$. The proof is provided in Appendix A.4. Figure 13 shows the experiments of the relative estimation errors of a *resnet18* model on *CIFAR10*, *CIFAR100* and *ImageNet* datasets. In our experiments, we empirically choose $K = 200$ due to this error upper bound.

## 5 Applications

We showcase multiple applications to demonstrate the potential uses of **I-ASIDE**[2]: (A1) Quantifying model global robustness by examining spectral importance distributions, (A2) understanding the learning behaviours of image models on the datasets with supervision noise, (A3) investigating the learning dynamics of models from the perspective of the evolution of feature representation robustness in optimization, (A4) understanding the adversarial vulnerability of models by examining spectral importance distributions and (A5) studying image model out-of-distribution (OOD) robustness.

---

[2]The core code implementation is provided in supplementary material.

### 5.1 Showcase A1: Quantifying model global robustness

We measure the robustness by examining the spectral importance distributions or the spectral importance scores.

**Spectral importance distribution**: The experiments in Figure 4 showcase the measured spectral distributions of multiple models with multiple datasets. The results show that the spectral importance of the learned feature representations of trained models is non-uniform. Furthermore, models trained on larger training datasets exhibit higher robustness.

**Numerical comparison**: The experiment (a) in Figure 8 showcases the application of numerically comparing model robustness. The results correlate with the adversarial perturbation experiments in Figure 11 in which we measure the prediction probability variations between clean samples and adversarial samples for given labels with un-targeted FGSM/PGD attacking (Szegedy et al., 2013; Moosavi-Dezfooli et al., 2016; Goodfellow et al., 2014; Madry et al., 2017). The result implies that the numbers of trainable parameters seem not to play a crucial aspect on model robustness.

**Visualizing by projection**: The experiment (b) in Figure 8 showcases the projection of spectral importance distributions. The projection is performed by using t-SNE (Hinton & Roweis, 2002).

### 5.2 Showcase A2: Understanding how models respond to supervision noise

The experiments in Figure 9 showcase the investigation of how models respond to various label noise levels in the training process. We create the noisy-label datasets from clean human-annotated *Caltech101* by randomly re-assigning a proportion of labels. We vary the proportions (label noise levels) from 0 to 1 with stride 0.1. We train *googlenet*, *resnet18* and *vit* on the derived noisy-label datasets with 70 epochs, learning rate 0.0025 and SGD optimizer. The image sizes are set to $64 \times 64$, $64 \times 64$ and $224 \times 224$ respectively. The results are visualized in 2D heat maps. We have observed the existence of a 'spectral drift' effect correlated with the levels of supervision signal noise in our preliminary showcase experiments. The models trained with lower levels of supervision signal noise tend to rely on the features dominated by low-frequency components, while the models trained with higher levels of supervision noise do not exhibit such a spectral preference. It is noteworthy that the 'spectral drift' effect is not a simple monotonic correlation with the levels of supervision noise. This phenomenon requires further investigation with additional experiments. Models exhibit disparate learning dynamics on human-annotated datasets and random-annotated datasets. As the theoretical analysis shown in Section 4.5, **I-ASIDE** is a quantification method for measuring the spectral importance of the variational mutual information between feature representations and labels. Intuitively, we surmise this '*spectral drift*' effect originates from that the supervision signals come from humans and the relevant features perceived by humans largely concentrate on low-frequency components due to the biological limit of our vision system. The training objectives for usual supervised classifiers are equivalent to estimating a variational lower bound of the mutual information between images and labels (See Section 4.5). Yet, it is not in the scope of this research.

### 5.3 Showcase A3: Investigating how feature representations are learned in training

The experiments in Figure 10 showcase how the feature representations learned by models evolve with respect to epochs in training process. The learning dynamics of models exhibit two stages. In the first stage, models readily learn more robust feature representations. In the second stage, models learn more non-robust feature representations to achieve higher accuracy.

We train '*googlenet*', '*efficientnet_v2_s*' and '*resnet18*' with 120 epochs, SGD optimizer and learning rate 0.0025. The training dataset is *Caltech*101. The spectral importance distributions are measured for every 5 epochs using **I-ASIDE**. We present the results using 2D heat maps with the corresponding training and validation loss curves. The first row is the training dynamics of feature robustness. The second row is the corresponding training and validation loss curve. The results suggest that models have two major learning stages: (1) Feature exploration stage and (2) feature exploitation stage. At the feature exploration stage (1), both training loss and validation loss drop synchronously. For example, the epochs 1 to 35 for *googlenet*, the

epochs 1 to 40 for *efficientnet_v2_s* and the epochs 1 to 70 for *resnet18*. This stage explores the features which are more general and the peaks of spectral distributions move towards low-frequency components. At the exploitation (2), models continue to refine the learned feature representations for further lower training loss. However, overfitting emerges at this stage. This stage uses more features dominated by low-frequency components. The results echo with the claim in a prior literature (Wang et al., 2020): High-frequency components are essential for the generalization capability of image models (Wang et al., 2020).

### 5.4 Showcase A4: Understanding adversarial vulnerability

Adversarial samples carry a high proportion of non-robust features (Su et al., 2018; Ilyas et al., 2019; Bai et al., 2021; Tsipras et al., 2018). Intuitively, the models with high spectral importance scores are less likely susceptible to adversarial perturbations. In Figure 11, we show how FGSM/PGD (Yuan et al., 2019; Madry et al., 2017; Akhtar & Mian, 2018; Chakraborty et al., 2018) adversarial perturbations correlate with the spectral importance scores by **I-ASIDE**. The full results are provided in supplementary material. We measure the prediction probability variations between the prediction probabilities of clean samples and adversarial samples. We use the 'smooth' version of Softmax function with temperature $\mathcal{T}$ to convert logits into probabilities (Hinton et al., 2015; Goodfellow et al., 2016; Jang et al., 2016).

**Experimental method**: Let $I$ be some clean image and $I^*$ be the corresponding adversarial sample. Let $\mathcal{P}(y|x) : x \mapsto [0,1]$ be some image classifier which outputs the probability with respect to some category $y$. We define the adversarial perturbation $\Delta \widetilde{\mathcal{P}}$ as $\mathbb{E}_{x,y \sim (\mathcal{X}, \mathcal{Y})} |\mathcal{P}(y|x) - \mathcal{P}(y|x^*)|$ over some dataset $(\mathcal{X}, \mathcal{Y})$ where $(x,y)$ denotes an image and the corresponding label and $x^*$ denotes adversarial sample for $x$. We plot the prediction perturbations against spectral importance scores.

The results in Figure 11 shows that the adversarial perturbations are negatively correlated with spectral importance scores. The outliers remind us: Spectral importance distributions can merely reflect one aspect of the robustness of models.

### 5.5 Showcase A5: Studying model out-of-distribution (OOD) robustness

We also showcase how the spectral importance scores by using **I-ASIDE** correlates with the OOD perturbations to further justify our methodology. The measurement method is the same as in Section 5.4. We choose two OOD perturbations: Gaussian ($\sigma = 0.1$) and blurring with Gaussian kernel ($kernel = 3 \times 3$). The preliminary results show that there exists a correlation between OOD perturbations and spectral importance scores.

## 6 Limitations

**I-ASIDE** provides a unique insight into the interpretability of image models from the feature robustness on spectral perspective. Yet, our methodology has several limitations: (1) Spectral perspective can merely reflect one aspect of the holistic view of model robustness, (2) the masking strategies in coalition filtering have yet to be investigated in future, (3) the choice of $\beta$ in summarizing model robustness scores and (4) the resolution of the measured spectral distributions for image models.

Regarding the limitation (1), for example, carefully crafted malicious adversarial perturbations on low-frequency components can fool neural networks (Luo et al., 2022; Liu et al., 2023; Maiya et al., 2021). This further implies the complexity of this research topic. Regarding the limitation (2), we have preliminarily discussed several masking strategies (See Figure 7 and Section 4.4). We need further investigation regarding how to choose the optimal masking strategy. Regarding the limitation (3), providing a better summarizing function ($S : \mathbb{R}^M \mapsto [0,1]$) needs further investigation in future work. Regarding the limitation (4), as the spectral energy follows power law alike distribution, measuring the expectations of spectral contributions on decisions on high-frequency components poses a challenge. As such, **I-ASIDE** does not use sampling based approach on spectral coalitions to avoid inaccurate results and thus suffers from the computation cost by $\mathcal{O}(2^M)$. Fortunately, we do not need high spectral resolution to analyze model robustness problem.

# 7 Conclusions

This research is motivated by the link between feature robustness and the non-uniformity of the spectral structures of natural images, in tandem with the insight that neural networks are parameterized non-linear signal filters. **I-ASIDE** investigates how the 'neural signal filter' responds to signals on spectrum. We formulate the global feature importance as a decomposition problem on spectrum using coalition game theory. We theoretically show that our approach decomposes the mutual information between feature representations and labels onto spectrum. Our work provides a unique insight into deep learning research and enables a considerable number of applications as we have demonstrated. We have conducted a large number of experiments to support our claims.

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

# A Appendix

## A.1 A brief proof on B.A. variational bound

*Proof.* Barber & Agakov shows a tight variational lower bound of mutual information (Barber & Agakov, 2004). Qin et al. also demonstrates that classifiers trained with cross-entropy loss estimate the mutual information between inputs and labels (Qin et al., 2021). The proof technique can be found in multiple literature (Poole et al., 2019; Barber & Agakov, 2004; Qin et al., 2021).

Let $(\mathcal{X}, \mathcal{Y})$ be some dataset. Assume $\mathcal{X} \times \mathcal{Y}$ is compact in $\mathbb{R}^d$ and all probability measures on which are absolutely continuous in terms of Lebesgue measure. Let $P(y|x)$ and $P(y)$ be the prior conditional distribution and the prior label distribution on the dataset. The mutual information between $\mathcal{X}$ and $\mathcal{Y}$ is bounded above the mutual information between the representations of $\mathcal{X}$ (denoted as $Q(\mathcal{X})$) and the labels for the image classifier $Q(y|x)$. Let $Q^*(y|x) \in \{T|T(y|x) : x \mapsto (0,1]\}$ be an arbitrary probability measure map. Starting from the definition of mutual information:

$$\mathbb{I}(\mathcal{X}; \mathcal{Y}) = \underset{x,y \sim (\mathcal{X}, \mathcal{Y})}{\mathbb{E}} \log \frac{P(y|x)}{P(y)} \geqslant \sup_{Q^*} \left\{ \underset{x,y \sim (\mathcal{X}, \mathcal{Y})}{\mathbb{E}} \log \frac{P(y|x)}{P(y)} \frac{Q^*(y|x)}{Q^*(y|x)} \right\} \tag{21}$$

$$= \sup_{Q^*} \left\{ \underset{x,y \sim (\mathcal{X}, \mathcal{Y})}{\mathbb{E}} \log Q^*(y|x) - \underset{x,y \sim (\mathcal{X}, \mathcal{Y})}{\mathbb{E}} \log P(y) + \underset{x,y \sim (\mathcal{X}, \mathcal{Y})}{\mathbb{E}} \log \frac{P(y|x)}{Q^*(y|x)} \right\} \tag{22}$$

$$\geqslant \sup_{Q^*} \underset{x,y \sim (\mathcal{X}, \mathcal{Y})}{\mathbb{E}} \log Q^*(y|x) + H(\mathcal{Y}) \tag{23}$$

$$= \sup_{Q^*} \underset{x,y \sim (\mathcal{X}, \mathcal{Y})}{\mathbb{E}} \log Q^*(y|x) + H(\mathcal{Y}) \stackrel{\text{def}}{=} \mathbb{I}_{BA}(Q(\mathcal{X}); \mathcal{Y}) \tag{24}$$

where $\underset{x,y \sim (\mathcal{X}, \mathcal{Y})}{\mathbb{E}} \log \frac{P(y|x)}{Q^*(y|x)} = \underset{x \sim \mathcal{X}}{\mathbb{E}} \{KL\left[P(y|x)||Q^*(y|x)\right]\} \geqslant 0$ since $P(x,y) = P(y|x)P(x)$ and the non-negativity property of Kullback–Leibler divergence (Gibbs' inequality) (Poole et al., 2019; Barber & Agakov, 2004; Qin et al., 2021; Kullback, 1997). The bound is tight *if and only if*: $Q^* = P$. Readers can refer the similar proof technique and the definition to the equation (2) in literature (Poole et al., 2019) and the equation (1) and the equation (3) in literature (Barber & Agakov, 2004).

$\square$

## A.2 Discrete Fourier Transform (DFT)

The notion 'frequency' measures how 'fast' the outputs can change if inputs vary. High frequency suggests small variations in inputs can cause large changes in outputs. In terms of images, the 'inputs' are the pixel spatial locations while the 'outputs' are the pixel values.

Let $x : (i, j) \mapsto \mathbb{R}$ be some 2D image with dimension $M \times N$ which sends every location $(i, j)$ to some real pixel value where $(i, j) \in [0, M - 1] \times [0, N - 1]$. Let $\hat{x} : (m, n) \mapsto \mathbb{C}$ be the Discrete Fourier transform (DFT) of $x$ in which $(m, n) \in [0, M - 1] \times [0, N - 1]$ denotes some 2D spatial frequency points. Let $\mathscr{F} : x \mapsto \hat{x}$ be some DFT functional operator. The DFT of $x$ is given by:

$$\hat{x}(m, n) := \mathscr{F}(x)(m, n) = \sum_{j=0}^{N-1} \sum_{i=0}^{M-1} x(i, j) e^{-\mathrm{i}2\pi(\frac{m}{M}i + \frac{n}{N}j)}. \tag{25}$$

Let $\mathscr{F}^{-1} : \hat{x} \mapsto x$ be the inverse DFT (IDFT) functional operator. The IDFT is given by:

$$x(i, j) := \mathscr{F}^{-1}(\hat{x})(i, j) = \frac{1}{M \cdot N} \sum_{n=0}^{N-1} \sum_{m=0}^{M-1} \hat{x}(m, n) e^{\mathrm{i}2\pi(\frac{i}{M}m + \frac{j}{N}n)}. \tag{26}$$

### A.3 Simplifying score formula

Let $(\mathcal{I}, v)$ be a spectral coalition game with the characteristic function $v$ with spectral player set $\mathcal{I}$. Set $\boldsymbol{\beta} := (\beta^k)_{k=0}^{M-1}$ where $\beta \in (0,1)$. Let $\Psi(v)$ be some spectral importance distribution. Set:

$$\Psi^*(v) = \frac{\Psi(v) - \min \Psi(v)}{||\Psi(v) - \min \Psi(v)||_1}. \tag{27}$$

We consider the inner product of $\boldsymbol{\beta}$ and and $\Psi^*(v)$ against the inner product of $\boldsymbol{\beta}$ and an equiprobability random decision to assess the decision:

$$\left| \boldsymbol{\beta}^T \Psi^*(v) - \boldsymbol{\beta}^T \frac{\mathbf{1}}{M} \right| = \left| \boldsymbol{\beta}^T \Psi^*(v) - \frac{||\boldsymbol{\beta}||_1}{M} \right|. \tag{28}$$

We normalize the above result and set:

$$S(v) := \frac{\left| \boldsymbol{\beta}^T \Psi^*(v) - \frac{||\boldsymbol{\beta}||_1}{M} \right|}{\sup \left| \boldsymbol{\beta}^T \Psi^*(v) - \frac{||\boldsymbol{\beta}||_1}{M} \right|} \tag{29}$$

$$= \frac{\left| \boldsymbol{\beta}^T \Psi^*(v) - \frac{||\boldsymbol{\beta}||_1}{M} \right|}{\sup \left| ||\boldsymbol{\beta}||_2 \cdot ||\Psi^*(v)||_2 - \frac{||\boldsymbol{\beta}||_1}{M} \right|} \tag{30}$$

$$= \frac{\left| \boldsymbol{\beta}^T \Psi^*(v) - \frac{||\boldsymbol{\beta}||_1}{M} \right|}{\left| ||\boldsymbol{\beta}||_2 - \frac{||\boldsymbol{\beta}||_1}{M} \right|} \tag{31}$$

$$= \left| \frac{\boldsymbol{\beta}^{*T} \Psi^*(v) - \frac{1}{M} \frac{||\boldsymbol{\beta}||_1}{||\boldsymbol{\beta}||_2}}{1 - \frac{1}{M} \frac{||\boldsymbol{\beta}||_1}{||\boldsymbol{\beta}||_2}} \right|. \tag{32}$$

where $\boldsymbol{\beta}^* = \frac{\boldsymbol{\beta}}{||\boldsymbol{\beta}||_2}$ and $\sup \left| \boldsymbol{\beta}^T \Psi^*(v) - \frac{||\boldsymbol{\beta}||_1}{M} \right|$ is derived by:

$$\sup \left| \boldsymbol{\beta}^T \Psi^*(v) - \frac{||\boldsymbol{\beta}||_1}{M} \right| = \left| \sup \boldsymbol{\beta}^T \Psi^*(v) - \frac{||\boldsymbol{\beta}||_1}{M} \right| \tag{33}$$

$$= \left| \sup ||\boldsymbol{\beta}||_2 \cdot ||\Psi^*(v)||_2 - \frac{||\boldsymbol{\beta}||_1}{M} \right| \quad \text{s.t. } ||\Psi^*(v)||_1 = 1 \tag{34}$$

$$= \left| ||\boldsymbol{\beta}||_2 - \frac{||\boldsymbol{\beta}||_1}{M} \right| \quad \text{since } ||\Psi^*(v)||_2^2 \leqslant ||\Psi^*(v)||_1^2. \tag{35}$$

Set $\eta = \frac{1}{M} \frac{||\boldsymbol{\beta}||_1}{||\boldsymbol{\beta}||_2}$:

$$S(v) = \left| \frac{\boldsymbol{\beta}^{*T} \Psi^*(v) - \eta}{1 - \eta} \right|. \tag{36}$$

$$Q.E.D.$$

### A.4 Proof of error bound

Let $K$ be the number of the samples of some baseline dataset. Let:

$$\Delta v(\tilde{\mathcal{I}}, \mathcal{I}_i) := v(\tilde{\mathcal{I}} \cup \{\mathcal{I}_i\}) - v(\tilde{\mathcal{I}}) \tag{37}$$

and

$$\Delta v(\mathcal{I}_i) := \left( \Delta v(\tilde{\mathcal{I}}, \mathcal{I}_i) \right)_{\tilde{\mathcal{I}} \subseteq \mathcal{I}} \tag{38}$$

and

$$W := \left( \frac{1}{M} \binom{M-1}{|\tilde{\mathcal{I}}|}^{-1} \right)_{\tilde{\mathcal{I}} \subseteq \mathcal{I}}. \tag{39}$$

Hence:

$$\psi_i(\mathcal{I}, v) = W^T \Delta v(\mathcal{I}_i) \tag{40}$$

where $||W||_1 \equiv 1$ since $W$ is a probability distribution. Let $\psi_i^*$, $\Delta v^*(\mathcal{I}_i)$ and $\Delta v^*(\tilde{\mathcal{I}}, \mathcal{I}_i)$ be estimations with $K$ samples using Monte Carlo sampling. The error bound with $\ell_1$ norm is given by:

$$\epsilon \overset{\text{def}}{=} \sup_i \mathbb{E}_{(\mathcal{X}, \mathcal{Y})} ||\psi_i^*(\mathcal{I}, v) - \psi_i(\mathcal{I}, v)||_1 = \sup_i \mathbb{E}_{(\mathcal{X}, \mathcal{Y})} ||W^T \Delta v^*(\mathcal{I}_i) - W^T \Delta v(\mathcal{I}_i)||_1 \tag{41}$$

$$\leqslant \sup_i \mathbb{E}_{(\mathcal{X}, \mathcal{Y})} ||W||_1 \cdot ||\Delta v^*(\mathcal{I}_i) - \Delta v(\mathcal{I}_i)||_\infty \qquad \left( \text{Hölder's inequality} \right) \tag{42}$$

$$= \sup_i \mathbb{E}_{(\mathcal{X}, \mathcal{Y})} || \sum_{\tilde{\mathcal{I}} \subseteq \mathcal{I} \setminus \mathcal{I}_i} \left( \Delta v^*(\tilde{\mathcal{I}}, \mathcal{I}_i) - \Delta v(\tilde{\mathcal{I}}, \mathcal{I}_i) \right) ||_\infty \tag{43}$$

$$\leqslant \sup_i \mathbb{E}_{(\mathcal{X}, \mathcal{Y})} 2^{M-1} \sup_{\tilde{\mathcal{I}}} ||\Delta v^*(\tilde{\mathcal{I}}, \mathcal{I}_i) - \Delta v(\tilde{\mathcal{I}}, \mathcal{I}_i)||_\infty \tag{44}$$

$$= \sup_i \mathbb{E}_{(\mathcal{X}, \mathcal{Y})} 2^{M-1} \sup_{\tilde{\mathcal{I}}} ||\Delta v^*(\tilde{\mathcal{I}}, \mathcal{I}_i) - \Delta v(\tilde{\mathcal{I}}, \mathcal{I}_i)||_1 \tag{45}$$

$$\leqslant 2^{M-1} \cdot \left\{ \frac{Var(\Delta v^*)}{K} \right\}^{\frac{1}{2}} \tag{46}$$

where $Var(\Delta v^*)$ gives the upper bound of the variance of $\Delta v^*(\tilde{\mathcal{I}}, \mathcal{I}_i)$.

## A.5 Full experimental data in training dynamics

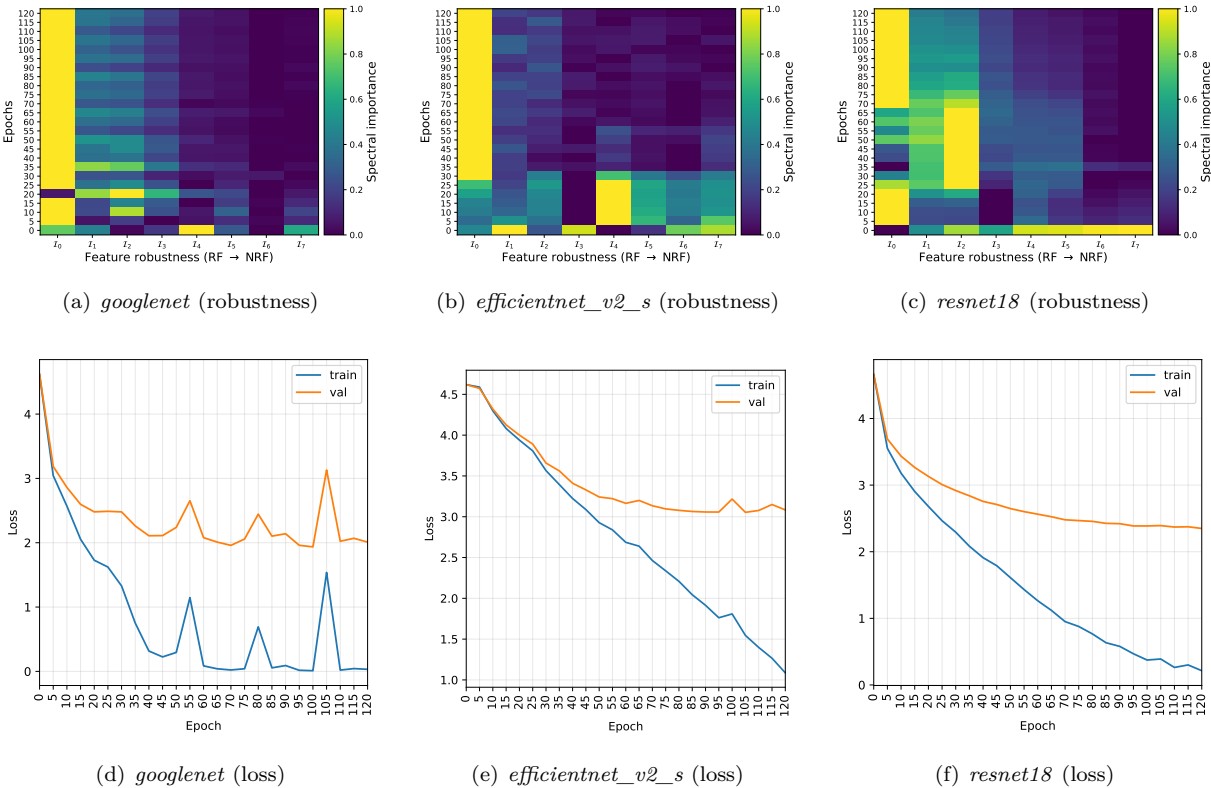

Figure 14: This is an application showcase (A3) in research demonstrating how the robustness of feature representations of models evolves with respect to training epochs. We train '*googlenet*', '*efficientnet_v2_s*' and '*resnet18*' with 120 epochs, SGD optimizer and learning rate 0.0025. The training dataset is *Caltech*101. The spectral importance distributions are measured for every 5 epochs using **I-ASIDE**. We present the results using 2D heat maps with the corresponding training and validation loss curves. The first row is the training dynamics of feature robustness. The second row is the corresponding training and validation loss curve. The results suggest that models have two major learning stages: (1) Feature exploration stage and (2) feature exploitation stage. At the feature exploration stage (1), both training loss and validation loss drop synchronously. For example, the epochs 1 to 35 for *googlenet*, the epochs 1 to 40 for *efficientnet_v2_s* and the epochs 1 to 70 for *resnet18*. This stage explores the features which are more general and the peaks of spectral distributions move towards low-frequency components. At the exploitation (2), models continue to refine the learned feature representations for further lower training loss. However, overfitting emerges at this stage. This stage uses more features dominated by low-frequency components. The results echo with the claim in a prior literature (Wang et al., 2020): High-frequency components are essential for the generalization capability of image models (Wang et al., 2020).

### A.6    Full experimental data in adversarial perturbations

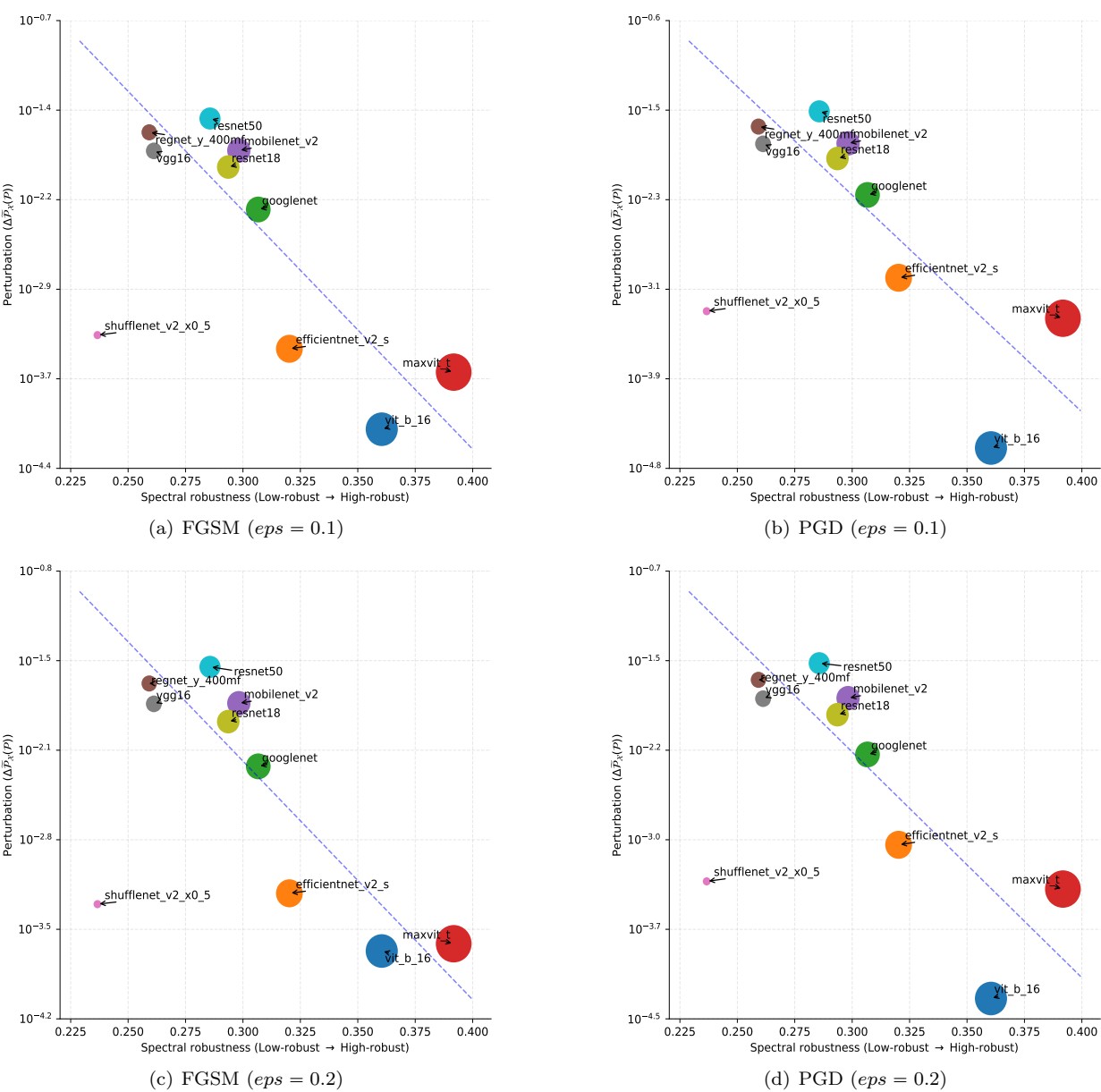

Figure 15: This is an application showcase (A4) in research demonstrating that there is a correlation between adversarial perturbations and the spectral importance scores using **I-ASIDE**. The adversarial perturbations are measured by the statistical expectations of prediction probability variations (See Section 5.4). We choose multiple pretrained models (on *ImageNet*) and perform un-targeted FGSM/PGD attacking with $eps = 0.1$ and $eps = 0.2$ respectively. The adversarial samples carry a high proportion of non-robust features (Ilyas et al., 2019) (See Figure 2). For most models, adversarial perturbations are negatively proportional to spectral importance scores. The circle sizes are proportional to spectral importance scores. It is also notable there are some outliers (e.g. *shufflenet*), and this implies the perturbation-based robustness research can merely capture an aspect of the holistic robustness of models.

