# OpenReview forum: "I-ASIDE: Towards the Global Interpretability of Image Model Robustness through the Lens of Axiomatic Spectral Importance Decomposition"
_TMLR — Rejected by TMLR_

### Review · Reviewer_K5tv · 2023-10-10

**Summary Of Contributions:**

The authors introduce I-ASIDE, a new method that computes spectral importance contributions of neural network-based image model decisions using coalitional game theory. This method first partitions the radial spectrum of an image into M equispaced regions, where each partition acts as a ‘spectral player’. The decomposition is then defined as the unique decomposition that satisfies a set of desirable axioms from coalitional game theory. The resulting spectral importance distribution is summarized into a spectral importance score using a predefined robustness prior, which quantifies the global model robustness as a scalar.

It is well-known that small spatial perturbations can have an outsized effect on network features dominated by high-frequency components. Hence, computing the spectral importance of high-frequency components with I-ASIDE can provide insights into the robustness of neural network models.

The authors apply the I-ASIDE method to investigate and interpret the robustness of image classifiers from a spectral importance perspective. Specifically, they find that:

1. Trained models exhibit highly non-uniform spectral importance distributions.
2. Models trained on larger datasets demonstrate less importance in high spectral regions.
3. The number of trainable parameters does not significantly affect the spectral importance score.
4. Higher label noise tends to result in increased importance of high spectral regions.
5. The learning dynamics of spectral importance distribution undergo two stages: robust features are learned before non-robust features.
5. Adversarial vulnerability correlates with spectral importance scores.

**Audience:**

Yes

**Broader Impact Concerns:**

I have no concerns on the ethical implications of the work that would require adding a Broader Impact Statement.

**Claims And Evidence:**

No

**Requested Changes:**

All of the requested changes are critical:


1. In the authors' statement "The results [in Figure 5] show that the models trained with higher label noise levels tend to use a higher proportion of non-robust features," I find it challenging to visually validate this claim from Figure 5 (a), (b), and (c). To strengthen this assertion, consider quantifying or enhancing the visualization of the result. After making this change, could the authors also indicate whether the claim is consistently true across varying label noise levels, i.e. does the importance of non-robust features gradually increase with increasing label noise strength?

2. The deduction from Figure 3 that "models trained on larger training datasets exhibit higher robustness" merits additional experimental substantiation. This should include averaging importance scores over different model runs and incorporating a broader range of dataset sizes. Consider plotting spectral importance distributions or scores across different fractions of the training dataset, like 25%, 50%, 75%, and 100% of datasets such as ImageNet, STL10, or CIFAR10/100.

3. The use of the robustness prior for the importance score was chosen to favor the importance of high spectral components. After line (8), the choice of $\beta=0.75$ is introduced. However, this scalar value for $\beta$ doesn't align with its earlier vectorial definition. Once this discrepancy is resolved, it would be helpful to offer a heuristic justification for the choice of the prior, aiding in reader comprehension.

4. On the topic of anisotropy: Anisotropy is the structural property of non-uniformity in different directions, as opposed to isotropy.  The authors frequently refer to the "spectral anisotropy of the spectral importance distribution" in the manuscript. My interpretation suggests that a better term  is the  "non-uniformity of the spectral importance distribution". The caption for Figure 1 further adds to this confusion. A clearer caption might be "This diagram illustrates the correlation between the power-law-like spectral density of natural images and the robustness of feature representations." If my interpretation of "spectral anisotropy" is incorrect, I would appreciate a detailed explanation.

5. Could the authors explain why the four Shapley axioms were adapted to a stronger version.

6. Could the authors correct the grammatical error in the following sentence: "This research is inspired by the power-law like spectral structures of natural images and feature robustness can be indexed by radial spectrum (See Figure 1)".

7. In Section 4.3, the authors say "This result gives an insight into designing better optimization goals by introducing robustness penalty towards more robust models". This was not corroborated by any experiments. Hence, I suggest either provide experiments for this claim or reduce the claim, e.g. reduce the claim to plans for future work.

8. The following sentence is very convoluted and would benefit from rewriting as well as a citation: "The fragility of learned features correlates with the energy distributions of spectral structures from the perspective of the susceptibility of feature representations to perturbations."

I hope the authors can address all my concerns. I think the submission would be of value to the community. If the authors address the above listed concerns I would to recommend acceptance. Until then, I don't see sufficient support for some important claims of the paper.

**Strengths And Weaknesses:**

## Strengths
1. Good Motivation: Computing global spectral importance scores is well motivated in light of the fact that small spatial perturbations can have an outsized effect on network features dominated by high-frequency components.
2. Well-founded Approach: Computing the spectral importance distribution via coalitional game theory gives the framework a solid basis.
3. Interesting Experimental Results: The authors compute the spectral importance distributions for different neural networks architectures, training settings, and training phases. The claimed results are interesting.


## Weaknesses
1. Soundness of Experimental Support: I am concerned about the experimental support for some claims of the paper.
	* The authors say "The results [in Figure 5]
show that the models trained with higher label noise levels tend to use a higher proportion of non-robust
features." Visually it is difficult for me to confirm this from Figure 5 (a), (b), and (c).
	* The authors deduce from Figure 3 that "models trained on larger training
datasets exhibit higher robustness." This is a strong interesting claim that I believe requires more experimental support. This should include averaging importance scores over model runs and more variation in dataset size. For example, one could plot importance distributions or scores for various fractions of training set size (e.g. 25%, 50%, 75%, 100% of imagenet, STL10, or CIFAR10/100).
2. Clarity: Generally, I found some parts of the paper require more clarity and elaboration. I will specify the parts in the requested changes section.

---

> ### Author Response · Authors · 2023-10-17
> **Thank you. We will update the manuscript as per requested changes on openreview shortly.**
>
> To Reviewer K5tv:
>
> We would like to thank the valuable and constructive comments on our work. The raised concerns will greatly improve the soundness of our research and the readability of our manuscript.
>
> We are now conducting the extra requested experiments, and revising the manuscripts as per the requested changes. We will update the manuscript on openreview shortly.
>
> Thank you.
>
> Paper1613 Authors

---

### Review · Reviewer_pLTm · 2023-10-20

**Summary Of Contributions:**

The submission explores the connection between global (i.e., over a population) spectral feature importance and robustness of deep-learning-based image classifiers. The authors propose a principled way to evaluate global importance of spectral features inspired by the axioms of the Shapley value. With this, they introduce a *spectral importance score* that quantifies how well the distribution of the importance of spectral features from low- to high-frequency aligns with a *robust prior* that favors low-frequency spectral features. The intuition is based on prior works which show convolutional neural networks are anisotropic in the Fourier basis, and that high-frequency spectral features are less robust than low-frequency ones.

Different case-studies showcase how the idea of spectral importance can be deployed in practice to study the behavior of deep-learning-based image classifiers across a variety of scenarios: from model training with label noise to robust evaluation. Empirical results show there exists a correlation between spectral importance score and adversarial robustness.

**Audience:**

Yes

**Claims And Evidence:**

No

**Requested Changes:**

**Prior works on global feature importance**

The idea of *explaining* the risk of a classifier with global feature attributions that satisfy the axioms of the Shapley value has been explored by Covert et al. in *``Understanding Global Feature Contributions With Additive Importance Measures''* (2020). In particular, Eq. (4) in the submission is a variation of Eq. (2) in the reference. The submission distinguishes itself by evaluating the importance of spectral features rather than pixels for image classifiers. It might be helpful to clarify how the two are related.

---

**Prior works on spectral feature importance**

From an information-theoretical perspective, Kolek et al. study *local* spectral feature importance in *``Cartoon Explanations of Image Classifiers''*. The submission is different in that feature importance is evaluated over a population rather than over an individual sample, and in the Fourier domain instead of the wavelet domain. Readers might appreciate this be clarified in the submission.

---

**Notation clarifications**

It is my understanding the gain measure $\mu: \tilde{\mathcal{I}} \to \mathbb{R}$ is what the game-theoretic literature commonly refers to as *characteristic function*. If that is the case, the gain measure should be the same for all subsets of $\mathcal{I}$, i.e. $\mu: \mathcal{P}(\mathcal{I}) \to \mathbb{R}$, where $\mathcal{P}(\mathcal{I})$ is the power set of $\mathcal{I}$.

I found the (recursive?) definition of $V_{\tilde{\mathcal{I}}}(\mu)$ on page 5 somewhat confusing because $\mu$ is never applied to any subset. This is what I implicitly thought of it: $V_{\tilde{\mathcal{I}}}(\mu) \coloneqq \left(\mu(\tilde{\mathcal{I}}_i)\right) _{i=0}^{...}$. Footnote 4 introduces boldface notation for vectors but that is never used.

In the decision risk function design, I am not sure I understand the motivation of using the *negative* prediction probability. This seems to trickle down to Eq. (5) where the sign is flipped again. Maybe this could be avoided?

Eq. (3) implicitly states all images have 3 channels, and there is no need for this assumption for the results to hold.

Similarly to above, the definition of $V_{\mathcal{X}}(R)$ does not seem to define what $V_{\mathcal{X},\mathcal{I}_i}(R)$ is. In the line below $V _\mathcal{X,i}(R)$ is used which I assume to be a typo.

Eq. (5) might be missing a transpose? $C(\mathcal{I})$ and $V_{\mathcal{X}}(R)$ are vectors are the RHS is a scalar. This is okay in Eq. (6) because $C$ is a matrix.

A few times it is mentioned *for some neural network $R: (I, y) \to \mathbb{R}$* which could lead to confusion between the risk function and the underlying neural network itself. A neural network does not have the true label as input. This may be a trivial distinction to expert readers but make the manuscript less accessible to a broader audience.

---

**Questions about axiomatic characterization**

*Strong* efficiency axiom: could the authors comment on the choice of the strong version of the efficiency axiom? I understand this axiom lets Eq. (5) go through smoothly, but if this is the only reason, one could also write the usual Shapley values in matrix form. Would anything else break with the usual efficiency characterization of the Shapley value? Or am I missing the point here.

Symmetry axiom: given two players $j,k \in [n] \coloneqq \\{1, \dots, n\\}$ and characteristic function $v$, the axiomatic characterization of the Shapley value requires that $\forall C \subseteq [n] \setminus \\{j, k\\}$, $v(C \cup \\{j\\}) = v(C \cup \\{k\\})$. The axiom is presented in the submission only for one spectral player coalition $\tilde{\mathcal{I}}$. Is this because of the strong efficiency axiom?

Nullity axiom: similarly, given $\mathcal{I}^*$, the axiom would usually be $\forall \tilde{\mathcal{I}} \subseteq [n] \setminus \\{\mathcal{I}^*\\}$, and not the other way around.

Besides these points, If the axiomatic characterization described above is equivalent to the Shapley value's, then $V_{\mathcal{X}}(R)$ is the vector containing the Shapley values of each spectral feature with respect to the game identified by the decision risk function. If this is true, then the presentation of the decomposition should reflect it. I am also confused because if this is indeed the case, then the pseudoinverse matrix presented in Appendix E does not reflect the weighting scheme of the Shapley value. For example, the rows of the matrix should sum up to 0, but this is not the case.

On the other hand, if the proposed decomposition is different from the Shapley value, then the axiomatic characterization cannot be the same as the Shapley value's because of its uniqueness. Differences should then be discussed and highlighted, and uniqueness of the solution concept that satisfies the (now different) proposed axiomatic characterization should be proved, or the text modified accordingly.

I am looking forward to discussing with the authors to clarify these doubts.

---

**Question about masking strategy**

Whenever applying a data-dependent function to some input with restricted features, it is important to devise some strategy to maintain the masked inputs close to the training distribution. In practical terms, I assume the neural networks used in the experiments were trained on images without any spectral filtering. This would render the images showed in Fig. 2 (center column) out-of-distribution for the neural network. Eq. (4) seems to adopt the strategy to zero features that are not in the spectral coalition of interest. Other strategies have been introduced in the explainability literature (e.g., unconditional expectation as presented in the original SHAP paper by Lundberg and Lee from 2017, we refer to the recent review by Chen et al. *"Algorithms to estimate Shapley value feature attributions"*).

Zero-ing features out might be a reasonable strategy in this setting, but the choice should be discussed in the manuscript.

---

**Questions about the spectral importance score**

I found the formulation of this score interesting! I wish the authors could spend some more time on Sec. 3.7 to introduce the score and its intuitive motivation, which I found quite nice.

Regarding the definition of $V_{\mathcal{X}}^*$, this seems to be a map between a vector in some domain (by the way, I am curious about the domain of $V_{\mathcal{X}}$, I think it should be bounded because the risk is bounded?) onto the simplex in $M$ dimensions. However, $V_{\mathcal{X}}^*$ is not the projection operator onto the simplex. I am wondering whether the authors could expand on how this particular choice of map was informed, and whether they expect empirical results to change depending on which map is used.

Similarly, I am curious about the choice of $\beta = 0.75$, I am not familiar with robustness literature and unaware whether this is a common choice.

---

**Estimation of error upper bound**

I appreciated this section: it is useful to see how the proposed sampling strategy behaves as the number of samples is increased.

---

**Minor assorted comments**

I wonder whether the authors expect results to be affected by the choice of spectral features. In the submission, $\ell_{\infty}$ balls are used, what about other norms?

Strong efficiency axiom definition: typo in *on coaltion* -> *in coalition* ?

Could the correlation lines be included in Figure 7? This might better showcase the trend at a first glance.

I think it is interesting vision transformers seem to be outliers in Figure 7, given the motivating work was for convolutional neural networks. Do the authors think something fundamentally different might be going on with those models?

Section 4.2: typo in *as\`label noise -> as \`label noise*

**Strengths And Weaknesses:**

Strenghts:
* The idea of deploying ideas of global feature importance to analyze the robustness of image classifiers is novel and intriguing.
* The experimental section does a good job at showcasing the practical applications of the proposed method.

Weaknesses:
* Notation---coming from the explainability field---is somewhat unusual and hard to follow
* Seminal works on global feature importance and spectral feature importance are missing

I am looking forward to engage with the authors to discuss a few comments and questions that came up while reading the submission.

---

> ### Author Response · Authors · 2023-10-21
> **We are currently working on revising this manuscript as per your concerns and requested changes.**
>
> To Reviewer pLTm:
>
> We would like to thank your time and the concerns raised in this manuscript. These concerns are valuable and can greatly improve the quality of our research. We appreciate your constructive comments once again. We are currently working on revising this manuscript as per your concerns and requested changes. We will online a new manuscript on OpenReview shortly.
>
> Thank you.
>
> Paper1613 Authors

---

> > ### Comment · Reviewer_pLTm · 2023-12-09
> > **Thank you for your revisions**
> >
> > I sincerely thank the authors for their efforts in addressing all questions and concerns.
> >
> > After reviewing the updated version of the manuscript, I still have some concerns regarding the presentation and soundness of the submission:
> >
> > 1. The updated "Global interpretability" paragraph in Sec. 2 is hand-wavy. In particular:
> > * The difference between local and global feature importance should be clearly presented.
> > * I am not sure I understand the claim that "the affine transforms (which transforms?) on images can change [...]" and how this showcases limitations of current explanation methods that evaluate feature importance in the pixel domain.
> > 2. The new "Mutual information in deep learning" paragraph in Sec. 2 is confusing. I am not sure I understand how it contributes to the presentation of the results. Several objects are presented without definition or motivation, for example:
> > * "enormous discriminators in GANS are variational divergence estimators [...]"
> > * "the $g_f(v)$
> > * "the corresponding conjugate function"
> > * Donsker-Varadhan Representation Theorem
> > 3. In the introductory paragraph of Sec. 3, I do not understand what the authors mean by "as a fairness utility distribution problem in the context of coalition game" and "decomposes [...] onto spectrum".
> > 4. In Sec. 3.1:
> > * $Q(y \mid x)$ is presented as a map into the unit interval $[0, 1]$. All results are presented for multi-class classification. I assume the authors mean they consider the logit for the ground-truth label. This should be made clear.
> > * The characteristic function is introduced as a function from the power set of the players into $\mathbb{R}$, and right after "the map $v$ is given as the form of the statistical expectation of some $Q(y \mid x)$. I think I understand what the authors mean, however, a classifier defined on a fixed domain cannot be plugged-in in place of a function defined on sets. This should be done more carefully.
> > * I do not understand what "the distant pixels" are or why they are easier to handle with $\ell_{\infty}$ norm
> > 5. In Sec. 3.2:
> > * The symmetry axiom still quite does not reflect standard definitions. In particular, I do not understand whether the authors mean $\forall \tilde{\mathcal{I}}:~\mathcal{I}_i,\mathcal{I}_j \notin \tilde{\mathcal{I}}$.
> > * The linearity axiom does not require $(u+v)(\tilde{\mathcal{I}}) = u(\tilde{\mathcal{I}}) + v(\mathcal{I})$: it requires that the attributions are linear in the games, not the individual characteristic functions.
> > * The dummy player axiom has not been updated with the new notation.
> > 6. Sec. 3.3:
> > * "We must implement spectral coalition sampling". Without explanation, this claim is confusing.
> > * A reference about planar comb filtering might be useful for readers not familiar with signal processing given the broad audience who might be interested in reading this work.
> > * I am not sure the "Replacement" strategy is appropriate given that it replaces the absent features with ones coming from a completely different image as shown in Figures 6(c) and (f). Usually, conditioning is needed in data-dependent sampling strategies to mask absent features.
> > 7. Sec. 3.4:
> > * It should be stated clearly whether Eq. (6) is a novel Theorem or a Lemma that follows from existing literature. The proof should be mentioned in the main text right after introducing the statement of the theorem/lemma. The proof itself should be moved to the Appendix.
> > * "the characteristic function measure the mutual information [...]". Does it measure this quantity or does it provide a lower bound?
> > 8. Sec. 3.5:
> > * "We set" and Eq. (14). I do not follow the motivation behind the randomly initialized weights baseline and approximating $\mathbb{E}[\Psi]$ with $1/M$. In the original version of the manuscript, I thought this was simply a baseline that assigned equal importance to all spectral features, which seemed reasonable.
> > * Similarly to Eq. (6), the statement of the upper bound should be clear and concise, and the proof moved to the Appendix.
> >
> > Miscellaneous comments:
> > 1. Footnote 2 is not necessary.
> > 2. There are several short sentences that begin with a citation and end with the same citation. "Nowozin et al theoretically show that enormous [...] they then propose [...] (Nowozin et al)". "Nowozin et al" is cited three times over 4 lines, which is quite redundant.
> > 3. "The correlation between the spectral density distributions of input features [...]" is too convoluted and should be rephrased.
> > 4. Figure 3 is referenced before Figure 2. Figure 4,5,6 are also referenced out of order on page 3.
> > 5. Figure 8:
> > * The colorbar is missing its label
> > * The caption should be made more concise. If the content of the caption is relevant for the discussion of the results, it should be moved to the main text. Otherwise, to an Appendix.
> > 6. Figure 9:
> > * Same consideration regarding caption.
> > * (d)(e)(f) seem more appropriate for an Appendix.

---

> > > ### Author Response · Authors · 2023-12-09
> > > **We appreciate your efforts and the great help once again here.**
> > >
> > > To Reviewer pLTm:
> > >
> > > We would like to sincerely thank your efforts of carefully examining the details in our manuscript. We agree with your concerns regarding our manuscript. The concerns are helpful for us to improve the research quality and soundness. We appreciate your efforts and the great help once again here.
> > >
> > > We are currently working on revising this manuscript as per your concerns and requested changes. We will online a new manuscript to address your concerns on OpenReview shortly.
> > >
> > > Thank you.
> > >
> > > Paper1613 Authors

---

> > > ### Author Response · Authors · 2024-01-05
> > > **The revisions and rebuttals as per the requests from the reviewer pLTm (part 1 round 2)**
> > >
> > > To Reviewer pLTm:
> > >
> > > We sincerely appreciate the huge help. The nice comments have great help for us to significantly improve the quality of the manuscript.
> > >
> > > We have finished the revisions as per your requests. We thank you again.
> > >
> > > We use $CiR2$ to denote the $i$-th concerns in the round 2.
> > >
> > > ## **C1R2: The updated "Global interpretability" paragraph in Sec. 2 is hand-wavy. In particular:**
> > >
> > > > - The difference between local and global feature importance should be clearly presented.
> > > > - I am not sure I understand the claim that "the affine transforms (which transforms?) on images can change [...]" and how this showcases limitations of current explanation methods that evaluate feature importance in the pixel domain.
> > >
> > > We much appreciate your concern. We agree with you regarding the clarification concern. We have revised the manuscript as per your requests. Please refer to the "**Global interpretability**" in the Section 3 "**Related work**".
> > >
> > > Regarding the the concern regarding "**affine transforms**", please refer to the depiction in the "**Global interpretability**" in the Section 3 "**Related work**":
> > > > ... Interpreting the global feature  importance in the spatial domain for images has inherited limits. For example, the showcase experiment of  literature (Covert et al., 2020) provides the global pixel-wise explanation of a multi-layer perceptron (MLP)  with the MNIST on spatial domain. The method assigns each pixel with an importance score and adds the  scores over MNIST to form the global interpretation. However, the positions and angles of the objects in  images can vary from image to image. Adding the pixel importance scores as global interpretation only offers  very limited information. We only read that models pay attentions in the centers of the images in MNIST  (Covert et al., 2020). Thus the interpretations in spatial domain often suffer from these limits and provide  very limited insights regarding robustness. Our work differs from these prior works in that we interpret global  model robustness in frequency domain. Frequency-domain transforms are not sensitive to the positions and  angles of the objects of images. For example, the spectrum of rotated or flipped images remains identical. ...
> > >
> > > We thank you again.
> > >
> > >
> > > ## **C2R2 The new "Mutual information in deep learning" paragraph in Sec. 2 is confusing....**
> > >
> > > > The new "Mutual information in deep learning" paragraph in Sec. 2 is confusing. I am not sure I understand how it contributes to the presentation of the results. Several objects are presented without definition or motivation, for example:
> > > > - "enormous discriminators in GANS are variational divergence estimators [...]"
> > > > - "the $g_f (v)$
> > > > - "the corresponding conjugate function" Donsker-Varadhan Representation Theorem
> > >
> > > We much appreciate the concern. We agree with your comments. We have revised the related work. In particular, we have overhauled the subsection "**mutual information**".
> > >
> > > We use the information theory in deep learning to conduct a theoretical analysis regarding the significance of the characteristic function $v$.  The analysis purports that our method itself is **theoretically interpretable**. Therefore, we provide a research background regarding this.
> > >
> > > We have changed the title of the "**Mutual information in deep learning**" into "**Information theory in deep learning**". We removed some irrelevant part and added motivations and analyses.
> > >
> > > Please kindly refer to the **Information theory in deep learning** in the Section 3 **Related work**. For example:
> > > > ... Despite the great success of deep learning, the question regarding how models represent input  features inside the black boxes is yet open to be answered. Tishby et al. answer this question by introducing  information bottleneck theory for interpreting the learning dynamics of neural networks (Tishby et al., 2000;  Saxe et al., 2019). The information bottleneck views models as an information encoding and decoding process.  Models learn to encode (extract) the information from inputs relevant to labels while neglect irrelevant  information (Saxe et al., 2019). Their work provides a motivation for  I-ASIDE. The mutual information  between the feature representations that models learn in training and labels reflect model performances. The  decomposition of such mutual information with respect to frequency can reflect the spectral contributions  to the learned information in black boxes. However, estimating the quantity of the mutual information is intractable since we do not have the knowledge regarding the joint distributions of feature representations  and labels. ...
> > >
> > > Thank you again.

---

> > > ### Author Response · Authors · 2024-01-05
> > > **The revisions and rebuttals as per the requests from the reviewer pLTm (part 3 round 2)**
> > >
> > > ## C4R2: "**... 2.  In Sec. 3.2: ...**"
> > > > The symmetry axiom still quite does not reflect standard definitions. In particular, I do not understand whether the authors mean ...
> > >
> > > We much appreciate the concern. We have revised the statement. Please refer to the Section 4.3 **Axioms and decomposition** in v9:
> > > > **Symmetry** Let $\widetilde{\mathcal{I}} \in 2^\mathcal{I}$ be some spectral player coalition. For $\forall~\mathcal{I}_i,\mathcal{I}_j \in \mathcal{I} \land \mathcal{I}_i,\mathcal{I}_j \notin \widetilde{\mathcal{I}}$, the statement $v(\widetilde{\mathcal{I}} \cup \{\mathcal{I}_i\}) = v(\widetilde{\mathcal{I}} \cup \{\mathcal{I}_j\})$ implies $\psi_i(\mathcal{I},v) = \psi_j(\mathcal{I},v)$. This axiom relabels the statement `**equal treatment of equals**' principle mathematically.
> > >
> > > Regarding the concern to **linearity**:
> > > > The linearity axiom does not require $(u+v)(\widetilde{\mathcal{I}})=u(\widetilde{\mathcal{I}})+v(\widetilde{\mathcal{I}})$ ...
> > >
> > > Please refer to literature [1] "**The Shapley value: Essays in honor  of Lloyd S. Shapley**":
> > > > ....Shapley defined a **$value$** for games to be a function that assigns to each game $v$ a number $\phi_i(v)$ for each $I$ in $U$. He proposed that such a function obey **three axioms**. ...
> > >
> > > **Comments**:
> > > > There are two equivalent axiom system to state the Shapley value theory: Three axioms or four axioms. But they are equivalent. We use latter. We have stated their relationship in our manuscript. Please refer to:
> > > >>... In the Shapley value theory literature (Roth, 1988), the efficiency axiom and the dummy player axiom are also relabeled as carrier axiom. ...
> > > > in the Section 4.3 "**Axioms and decomposition**"
> > >
> > > in the page 13 of the **additivity** axiom in the Chapter **Introduction to the Shapley value**:
> > > > ....
> > > > The third axiom, now called the **additivity** axiom, requires that, for any games $v$ and $w$
> > >
> > > **Comments**: **$v$ and $w$ are characteristic functions**
> > > For some game $(v, N)$ and game $(w, N)$ where $N$ is player set. For example, regarding the the Shapley Value for a player $i \in N$ in game $(v, N)$ is defined:
> > > $$
> > > \phi_i(v) : v,i \mapsto \mathbb{R}
> > > $$
> > > which assigns the player $i$ a real number.
> > >
> > > Continuing in the literature:
> > > > , $\phi(v) + \phi(w) = \phi(v + w)$ (i.e., $\phi_i(v) + \phi_i(w)=\phi_i(v+w)$ ...
> > > >...
> > > >This axiom, which specifies how the values of **different games** must be related to one another, is the driving  force behind Shapley's demonstration that there is a unique function $\phi$ defined on the space of all games that satisfies these three axioms.
> > >
> > > **Comments**: **This axiom guarantees the unique solution of Shapley Value.**
> > >
> > > References:
> > > - [1]. The Shapley value: Essays in honor  of Lloyd S. Shapley.
> > >
> > > Regarding this concern:
> > > > The dummy player axiom has not been updated with the new notation.
> > >
> > > We have revised this issue. Please refer to the Section 4.3 **Axioms and decomposition** in v9:
> > > > **Dummy player axiom**:
> > > > A dummy player (null player) $\mathcal{I}_{*}$ is the player who has no contribution such that: $\psi_*(\mathcal{I},v)=0$ and $v(\widetilde{\mathcal{I}} \cup \{\mathcal{I}_*\}) \equiv v(\widetilde{\mathcal{I}})$ for $\forall~\mathcal{I}_* \notin \widetilde{\mathcal{I}} \land \mathcal{I}_* \subseteq \mathcal{I}$.
> > >
> > > Sincerely thank you again.

---

> > > ### Author Response · Authors · 2024-01-05
> > > **The revisions and rebuttals as per the requests from the reviewer pLTm (part 6 round 2)**
> > >
> > > ## C6.2R2: "**... 2.  In Sec. 3.4: ...**"
> > >
> > > We sincerely thank you for raising the concern. We were also thinking of how to present the idea in a clear way. We have overhauled the Section 4.5 "**Characteristic function design and characterization**" to address your concern.
> > >
> > > To completely address such a concern, we have clarified the aspects in multiple places in new manuscript:
> > >
> > > - 1. What are features and **feature representations** in Section 1. Please refer to:
> > > > The term use of ‘feature’ is ambiguous and nuanced in deep learning community. In this research, we clarify  and differ the term into: (1) Input features (pixel features) and the representations of input features (**feature  representations**). The pixels of images are referred as ‘input features’ or ‘pixel features’. The ‘input features’  are fed into image models and the models represent the features inside. **The way that models represent the  input features is referred as ‘feature representations’.** Empirically, we can ‘observe’ how models represent  the input features by looking into the outputted probability distributions. ...
> > >
> > >
> > > - 2. What are the feature representations of $\mathcal{X}$ for given model $Q$. Please refer to Section 4.1 **Notations and preliminaries**:
> > > > We use the convention from topology (analysis) to denote the collection of the distributions $Q(y|x; \theta)$ over a set $\mathcal{X}$ as:
> > > > $$
> > > > Q(\mathcal{X}; \theta) \stackrel{def}{=}  \{Q(x; \theta) | x \in \mathcal{X} \}
> > > > $$
> > > > where $\theta$ denotes the parameter of classifier $Q$. Since the distributions of $Q$ over the dataset $\mathcal{X}$ represent the representations of the data in the dataset, we also call $Q(\mathcal{X}; \theta)$ as the representations of set $\mathcal{X}$. We ignore the parameter $\theta$ in all notations to maintain depiction succinct. For example, we denote $Q(y|x; \theta)$ as $Q(y|x)$, and $Q(x; \theta)$ as $Q(x)$ and $Q(\mathcal{X}; \theta)$ as $Q(\mathcal{X})$.
> > >
> > > - 3. Why $v$ can measure the mutual information between representations and labels. To answer this, we prove two claims:
> > > 	 - Claim 1: An image classifier $Q$ trained with usual cross-entropy loss estimate a variational lower bound of the mutual information between images $\mathcal{X}$ and labels $\mathcal{Y}$;
> > > 	 - Claim 2: The map $v$ defined on the top of the $Q$ and $(\mathcal{X}, \mathcal{Y})$ measures the mutual information between the representations $Q(\mathcal{X})$ and labels $\mathcal{Y}$ which also is a variational bound of the mutual information between images $\mathcal{X}$ and labels $\mathcal{Y}$.
> > >
> > > Please refer to the revised Section 4.5 **Characteristic function design and characterization**.
> > >
> > >
> > > ## C7R2: "**... Sec. 3.5: ...**"
> > >
> > > We appreciate the concerns. We have revised the manuscript to address the concerns.
> > >
> > > Regarding:
> > > > -  "We set" and Eq. (14). I do not follow the motivation behind the randomly initialized weights baseline and approximating $\mathbb{E}[\Psi]$ with $1/M$. In the original version of the manuscript, I thought this was simply a baseline that assigned equal importance to all spectral features, which seemed reasonable.
> > >
> > > In the manuscript v8, the intent was to give the thinking process. In fact, the $\frac{1}{M}$ directly from the **motivation** experiments in Figure 4 (manuscript v9). Please see the highlighted results with dashed **blue box** in Figure 4. The results show that models with randomised weights show **uniformity** in spectral importance. Please refer to:
> > > > The approximation equation (18) holds true because the motivation experiments in Figure 4 show that the measured spectral importance distributions of models with randomized weights exhibit uniformity.
> > >
> > > in the revised manuscript of the Section 4.6  **Summarizing spectral importance distribution**.
> > >
> > > Mathematically, the results in the motivation experiments (Figure 4) are:
> > > $$
> > > \mathbb{E}_{\widetilde{v}} [ \Psi^*(\widetilde{v})] \approx \frac{\mathbb{1}}{M}.
> > > $$
> > >
> > > The weight **randomising** with pytorch for model $Q$ is implemented with:
> > > ```python
> > > for n, p in Q.named_parameters():
> > >     nn.init.normal_(p)
> > > ```
> > >
> > > Please refer to **Figure 4** and the Section 4.6  **Summarizing spectral importance distribution**.
> > >
> > > > -  Similarly to Eq. (6), the statement of the upper bound should be clear and concise, and the proof moved to the Appendix.
> > >
> > > We have revised the manuscript as per your requests. Please refer to the Section 4.5 **Characteristic function design and characterization** in v9. We have restructured the representation of this section to deliver the presentation in a more logic way. We also moved the proof(s) into Appendix.
> > >
> > > Thank you again.

---

> > > ### Author Response · Authors · 2024-01-05
> > > **The revisions and rebuttals as per the requests from the reviewer pLTm (part 7 round 2)**
> > >
> > > ## C8R2: "**...  Miscellaneous comments: ...**"
> > >
> > > We much appreciate the concerns. We have revised the manuscript regarding your requests.
> > >
> > > Regarding:
> > > > - Footnote 2 is not necessary.
> > >
> > > We have removed the footnote 2.
> > >
> > > Regarding:
> > > > -  There are several short sentences that begin with a citation and end with the same citation. "Nowozin et al theoretically show that enormous [...] they then propose [...] (Nowozin et al)"....
> > >
> > > We have revised the manuscript and rephrased the content. We will also continue to improve the manuscript until final version. We appreciate the concern again.
> > >
> > > Regarding:
> > > > - "The correlation between the spectral density distributions of input features [...]" is too convoluted and should be rephrased.
> > >
> > > We have revised the manuscript in v9.
> > >
> > > Regarding:
> > > > - Figure 3 is referenced before Figure 2. Figure 4,5,6 are also referenced out of order on page 3.
> > >
> > > We have adjusted some figures according to the order in the **main** representation flow. Although some figures are referenced earlier such as in Section 1.1 **Contributions**, the **main presentation** flow unfolds differently. We in particular design the order of the pictures in the **main presentation** flow. For example, we present the **motivation** experiments beforehand. We appreciate the concern. This reminds us to continue to optimise the figures in the presentation. We will continue to optimise.
> > >
> > > Regarding:
> > > > - 1.  Figure 8:
> > > >   - The colorbar is missing its label
> > > >   - The caption should be made more concise. If the content of the caption is relevant for the discussion of the results, it should be moved to the main text. Otherwise, to an Appendix.
> > >
> > > We have addressed this concern in two ways in the new manuscript v9:
> > > - If the spaces are allowed, we add color bar  titles into the figures directly.
> > > - Otherwise, if the spaces are not allowed as there are many subfigures in some figures. We instead state **the colorbars in the figures indicate frequency-domain magnitudes which are normalized into $[0,1]$** in captions.
> > >
> > > We appreciate this concern. We will continue to optimise the figures. Some figures are generated during the running of experiments. Some figures are created by using the data collected during the experiments. We will re-setup experiments (which need huge amount of time in particular for some experiments) to re-generate the figures in future.
> > >
> > > Regarding:
> > > > Figure 9:
> > > > - Same consideration regarding caption.
> > > > - (d)(e)(f) seem more appropriate for an Appendix.
> > >
> > > To completely address this concern in tandem with similar concerns, we have significantly shortened the captions. We also moved some figures such Figure 9 (d)-(f) to appendix as supplementary material.
> > >
> > > We sincerely appreciate your efforts and help.
> > >
> > > Thank you again.
> > >
> > > Authors.

---

> ### Author Response · Authors · 2024-01-05
> **The revisions and rebuttals as per the requests from the reviewer pLTm (part 2 round 2)**
>
> ## The term "fairness".
>
> > In the introductory paragraph of Sec. 3, I do not understand what the authors mean by "as a fairness utility distribution problem in the context of coalition game" and "decomposes [...] onto spectrum".
>
> The term **fairness** is a term in coalition game theory. We have revised the manuscript to address your concern. Please refer to the Section 4 "**Axiomatic spectral importance decomposition**":
> > ... The fairness distribution (decomposition or division) is a problem in coalition game theory and refers to divide the payoffs among players in a coalitional game in which the division must satisfy a set of axioms (Aumann & Maschler, 1985; Yaari & Bar-Hillel, 1984; Aumann & Dombb, 2015; Hart, 1989; Roth, 1988). ...
>
>
> and the Section 4.3 "**Axioms and decomposition**"
>
> > ... Dividing the ‘payoffs’ amid players in coalition games can exist multiple non-unique schemes. A fair division scheme is said that the scheme satisfies a set of desirable fairness axioms (Moulin, 1992; Roth, 1988; Hart, 1989; Winter, 2002; Shapley & Shubik, 1969; van den Brink, 2002). Shapley value theory is a solution concept which uniquely satisfies a set of desirable fairness axioms. We formulate the spectral importance decomposition as a fairness division problem in ‘spectral coalition game’. We state the essential axioms below. ...
>
> Intuitively, being **fair** must satisfy a set of axioms. For example, the "**labels**" or "**names**" should have no effect on payoff allocation schemes. The **symmetry axiom** guarantees that the **names** have no effects on allocation schemes.
>
>
>
>
>
>
> ## **C3R2: In Sec. 3.1: **
> > - $Q(y|x)$ is presented as a map into the unit interval $[0, 1]$. All results are presented for multi-class classification. I assume the authors mean they consider the logit for the ground-truth label. This should be made clear.
>
> We appreciate your concern. Actually, we adopt many notation conventions from topology and functional analysis in pure math. We have added a subsection "**Notations and preliminaries**" to clarify the notations.
>
> > - The characteristic function is introduced as a function from the power set of the players into R, and right after "the map v is given as the form of the statistical expectation of some Q(y ∣ x). I think I understand what the authors mean, however, a classifier defined on a fixed domain cannot be plugged-in in place of a function defined on sets. This should be done more carefully.
>
> For example, in some literature, a classifier is often denoted as:
> $$
> S_y(x;\theta): x \mapsto \mathbb{R}
> $$
> where $y$ denotes a given class. Unfortunately, this expression does not make sense to develop theoretical analysis in particular in **Bayesian school** and **information theory**. We need to express this in a way of statistics **friendly**:
> $$
> Q(y|x;\theta) \stackrel{def}{=} softmax(S_y(x;\theta)).
> $$
>
> Thus:
> $$
> Q(y|x; \theta) \stackrel{def}{=} \frac{e^{S_y(x; \theta)}}{\sum\limits_{y' \in \mathcal{Y}} e^{S_{y'}(x; \theta)}}.
> $$
>
> Clearly, $Q(y|x; \theta) \in [0,1]$ a probability measure which sends each $x$ to a probability for given $y$. We can further define probability distribution regarding the collection of $y$ (i.e. $\mathcal{Y}$):
> $$
> Q(x; \theta) \stackrel{def}{=} \biggl( Q(y|x; \theta) \biggr)_{y \in \mathcal{Y}}
> $$
> which collects the probability for each single class $y \in \mathcal{Y}$ and forms a distribution regarding $\mathcal{Y}$.
>
> Using the convention from analysis (and topology), we can define the collection $Q(x)$ over a set $\mathcal{X}$ (i.e. images):
> $$
> Q(\mathcal{X}; \theta) \stackrel{def}{=}  \{Q(x; \theta) | x \in \mathcal{X} \}
> $$
>
> ## **C3.1R2: In Sec. 3.1: **
> > 0.  I do not understand what "the distant pixels" are or why they are easier to handle with $\ell_\infty$ norm ...
>
> We appreciate this concern. To address this, we added the Figure 5 to address your concern. Please refer to the Figure 5 in manuscript v9.

---

> ### Author Response · Authors · 2024-01-05
> **The revisions and rebuttals as per the requests from the reviewer pLTm (part 4 round 2)**
>
> ## C5R2: "**... 2.  In Sec. 3.3: ...**"
> We appreciate the concerns. We have revised the manuscript and addressed the concerns.
>
> Regarding:
> > - "We must implement spectral coalition sampling". Without explanation, this claim is confusing.
>
> Please refer to:
> > To implement the spectral coalitions in the spectral coalition game defined in Section 3.2, we must implement  spectral coalition sampling. We implement the spectral coalition with ideal radial multi-band-pass digital signal filtering (Oppenheim, 1978; Roberts & Mullis, 1987; Pei & Tseng, 1998). The multi-band-pass digital  signal filtering refers that the signal components in pass-bands are preserved while the signal components in  stop-bands are suppressed. We state the implementation below.
>
> in the Section 4.4  **Spectral coalition filtering**.
>
> Regarding:
> > - A reference about planar comb filtering might be useful for readers not familiar with signal processing given the broad audience who might be interested in reading ...
>
> We have changed the term comb filter to multi-band-pass filter and added references and explanation. Please refer to:
> > ... We take some spectral  coalition $\widetilde{\mathcal{I}} \in 2^{\mathcal{I}}$ and crop the frequency components not present in  rI  by using a channel-wise plane multi-band-pass filter on frequency-domain (Oppenheim, 1978; Roberts & Mullis, 1987; Castleman, 1996; Jähne,  2005). The multi-band-pass mask matrices of allowing pass-bands and suppressing stop-bands are referred as  ‘transfer functions’ or ‘filters’ in the literature of digital signal processing. ...
>
> in the Section 4.4  **Spectral coalition filtering**. We may add more extra content in supplementary material in final version to introduce the **multi-band-pass** filtering in images.
>
>
> ## The term **baseline** and **absence assignment** in attribution analysis
>
> Regarding:
> >  I am not sure the "Replacement" strategy is appropriate given that it replaces the absent features with ones coming from a completely different image as shown in Figures 6(c) and (f). Usually, conditioning is needed in data-dependent sampling strategies to mask absent features.
>
> Please refer to:
> > **Absence baseline**: The term ‘**baseline**’ in **attribution analysis** context refers to the absence assignments of  players (Sundararajan et al., 2017; Shrikumar et al., 2017; Binder et al., 2016). For example, if we use ‘zeros’  to represent the absence of players, the ‘zeros’ are dubbed as ‘baseline’. Intuitively, ‘baseline’ is the ‘pretext’  for being ‘absent’ in the coalition game (Sundararajan et al., 2017).
>
> in the Section 4.4  **Spectral coalition filtering**. The "replacement" serves as some "**baseline**" referred in related literature in **attribution analysis** context.
>
> Intuitively, when we refer "attribution" or "contribution", there implies "variation" relative to "**baseline**" which is the setting we compare to. For example, suppose a **cat** presenting in **background**, we may say the **baseline** is the **background**.

---

> ### Author Response · Authors · 2024-01-05
> **The revisions and rebuttals as per the requests from the reviewer pLTm (part 5 round 2)**
>
> ## C6.1R2: "**... 2.  In Sec. 3.4: ...**"
> We appreciate the concerns. We have revised the manuscript and addressed the concerns.
>
> Regarding:
> > - It should be stated clearly whether Eq. (6) is a novel Theorem or a Lemma that follows from existing literature. The proof should be mentioned in the main text right after introducing the statement of the theorem/lemma. The proof itself should be moved to the Appendix.
>
> We have revised the manuscript. The **similar** results are in prior literature. However, there have **many faces** regarding the results which depend on the setting. In our setting, we need to express it to fit our setting. We use them a lemma. Please refer to:
> > Lemma **B.A. Feature Representation Variational Mutual Information Bound**. Let $(\mathcal{X}, \mathcal{Y})$ be some dataset. Let $Q^*(y|x)$ be some classifier. Let $Q$ be the optimal model optimized on the dataset with cross-entropy loss. Let $Q(\mathcal{X})$ be the representations of the set $\mathcal{X}$. Let $H(\mathcal{Y})$ be the differential entropy of $\mathcal{Y}$. Let $\mathbb{I}(\mathcal{X}; \mathcal{Y})$ is the mutual information between images and labels. The below variational bound holds and we refer the LHS as '**B.A. feature representation variational mutual information bound**' or '**B.A. mutual information bound**'. The bound is:
> $$
>     \mathbb{I}_{BA}(Q(\mathcal{X}); \mathcal{Y}) \stackrel{\text{def}}{=} \sup_{Q^*} \mathop{\mathbb{E}}\limits_{x,y \thicksim (\mathcal{X}, \mathcal{Y})} \log Q^*(y|x) + H(\mathcal{Y}) \leq \mathbb{I}(\mathcal{X}; \mathcal{Y})
> $$
> where the notation $\mathbb{I}_{BA}(Q(\mathcal{X}); \mathcal{Y})$ is read as: **The B.A. variational mutual information bound of dataset $(\mathcal{X}, \mathcal{Y})$ on classifier $Q$**.
>
> in the Section 4.5 **Characteristic function design and characterization**. We also revised the presentation in better organised way.
>
> Regarding:
> > - "the characteristic function measure the mutual information [...]". Does it measure this quantity or does it provide a lower bound?
>
> Considering the definition of the $v$:
> > Let $Q(y|x): x \mapsto [0, 1]$ be some image classifier. For a sample $x$, $Q(y|x)$ sends the $x$ to a discrete category probability distribution for $y \in \mathcal{Y}$. Let $Q(\mathcal{X})$ denote the representations of image set $\mathcal{X}$ with given classifier $Q$. We design the characteristic function \textcolor{blue}{on the classifier $Q$ over dataset $\mathcal{D}:=(\mathcal{X}, \mathcal{Y})$} as:
> > $$
> > v(\widetilde{\mathcal{I}}; Q, \mathcal{D}) := \mathop{\mathbb{E}}\limits_{x,y \thicksim (\mathcal{X}, \mathcal{Y})} \left\\{ \log Q(y | x \star \widetilde{\mathcal{I}}) - \log Q(y | x \star \emptyset)\right\\}
> > $$
> > where \textcolor{blue}{$\forall \widetilde{\mathcal{I}} \in 2^{\mathcal{I}}$. The characteristic function $v$ is a function of spectral coalitions ($\widetilde{\mathcal{I}} \in 2^{\mathcal{I}}$) over given dataset $(\mathcal{X}, \mathcal{Y})$ and classifier $Q$. The map $v$ also satisfies $v(\emptyset) = 0$ if the absences of spectral players are assigned to zeros in coalition filtering. We denote $v(\widetilde{\mathcal{I}}; Q, \mathcal{D})$ as $v(\widetilde{\mathcal{I}})$ to maintain depiction succinct.
>
> and
>
> >...
> >We now can show `Claim 2' holds. Applying the **Lemma 4.1. B.A. Feature Representation Variational Mutual Information Bound** into the equation, we can rewrite the characteristic function $v$ as:
>
>
> $$
> v(\widetilde{\mathcal{I}}) := \mathbb{I}_{BA}(Q(\mathcal{X} \star \widetilde{\mathcal{I}}), \mathcal{Y})
> $$
>
> $$
> -\mathbb{I}_{BA}(Q(\mathcal{X} \star \emptyset), \mathcal{Y})
> $$
>
> >which clearly measures the mutual information between between representations and labels, and also the B.A. variational bound of the mutual information between images and labels.
> >...
>
> We can thus say:
> - A. $v$ measures a **bound** of the **mutual information** between $\mathcal{X}$ and $\mathcal{Y}$. Please note: This statement has nothing to do with **classifier**.
> - B. $v$ measures the **mutual information** between $Q(\mathcal{X})$ and $\mathcal{Y}$. This **mutual information** is the **mutual information** between the representations of $\mathcal{X}$ and $\mathcal{Y}$ for given classifier $Q$.

---

### Review · Reviewer_LUjU · 2023-12-04

**Summary Of Contributions:**

This article presents a method to estimate a score for global interpretability of image model robustness, through image modifications in Fourier domain.The problem is formalized as a coalition game where the Shapely value theory is applied to derive the score. Numerical results show the correlation of this score with respect to properties of various neural networks, such as label noise / adversarial / out-of-distribution robustness.

**Audience:**

Yes

**Broader Impact Concerns:**

The idea of computing a global score to measure image model robustness is quite ambitious. Given the existing literature, it is still not very clear what is the main advantage of using this method, besides the claim that the proposed method seems to be a quite general tool. The connection between the proposed score and the intrinsic property of the studied models would require further study.

**Claims And Evidence:**

No

**Requested Changes:**

Clarify several definitions
* How do you define the value of v(tilde I union { I_i }) in eq. 1? In eq. 3, you used the convolution of x with tilde I, then how is this computed for tilde I union { I_i } ?
* The definition of Q in eq. 5 shows that it is an optimal classifier for a given model and dataset. When you use a perturbed dataset Q*tilde I in eq. 8, are you considering another optimal classifier for the perturbed dataset? The notation Q(X) is rather confusing as it is called the representations, then it becomes the optimal classifier.
* It is not clear how can Q be related to define a classifier with random weights in eq. 13. What is the definition of tilde v ?Why it is 1/M in eq. 14?

Clarify several contributions
* Introduction: when you talk about how features are represented inside black box models. What do the features mean? This point is not so clear from the rest of the article.
* The authors mentioned in the introduction that interpreting the global feature important in spatial domain has inherited limits. Is it possible to illustrate some of these limitations in your applications, to compare with your proposed method?
* The sensitivity analysis regarding the choice of hyper-parameters in the proposed approach can also added. In particular, would this explain why there is an outlier in the score correlation for the shufflenet in Fig 10? What happens if the eps is not set to 0.1.
* How the errors in Figure 12 are related to the eps in eq. 20?

**Strengths And Weaknesses:**

The idea of using Shapely value theory seems to be novel in the context of global interpretability. It was proposed in Lundberg and Lee 2017 to interpret feature importance. This article applies this theory by using considering spectral domain information (Fourier spectrum) of images.

Although the idea is interesting, the article should be improved regarding its contribution, as well its clarity.

---

> ### Author Response · Authors · 2023-12-09
> **Thanks for your valuable comments and the concerns raised regarding our work.**
>
> To Reviewer LUjU:
>
> We would like to thank your efforts and the concerns raised in this manuscript. These concerns are valuable and can greatly improve the quality of our research. We appreciate your constructive comments once again.
>
> We are currently working on revising this manuscript as per your concerns and requested changes. We will online a new manuscript to address the concerns on OpenReview shortly.
>
> Thank you.
>
> Paper1613 Authors

---

> > ### Comment · Reviewer_LUjU · 2024-01-05
> > **clarification**
> >
> > Dear authors,
> > Could you provide a more detailed or point-to-point answer to my questions ? I am not quite sure if you have addressed my concerns regarding the mathematical notations in coalition game theory and math context (e.g. How do you define the value of v(tilde I union { I_i }) in eq. 1?)

---

> ### Author Response · Authors · 2024-01-05
> **We apologise our delays.**
>
> To Reviewer LUjU,
>
> We sincerely thank you for your nice comments in our manuscript.
>
> We apologise our delay. Due to the Christmas, some our authors are in holiday.
>
> We have revised the manuscript as your requests and prepared the replies. We are on-lining the revised version and replies.
>
> Thank you again.
>
> Authors.

---

> ### Author Response · Authors · 2024-01-05
> **The revisions and rebuttals as per the requests from the reviewer LUjU (part 1)**
>
> We sincerely appreciate the huge help. We are working on the improvement of the manuscript in terms of soundness, correctness and readability. The nice comments greatly enable us to significantly improve the quality of the manuscript.
>
> We have finished the revisions as per your requests. We thank you again.
>
> We use $Ci$ to denote the $i$-th concerns.
>
>
> ## **C1: Regarding contributions.**
>
> We much appreciate the concern as per clarifying the contributions. We have revised the manuscript as per your request to clarify the contributions. To sum up, this work mainly aims to propose a quantification method of measuring the spectral importance of models. Although we have demonstrated the potential applications, the showcases are not in our main scope.
>
> We claim the contributions as:
> - **We introduce a method of measuring the spectral importance of image models**;
> - **We justify the proposed method from the perspective of information theory and coalition game theory;**
> - **We demonstrate the potential of our method in deep learning community to understand global feature representation robustness.**
>
> Please refer to the highlighted context in Section 2 "Contributions" accordingly.
>
>
> ## **C2:How do you define the value of $v(\widetilde{\mathcal{I}} \cup \{\mathcal{I}_i\})$ in eq. 1 (of manuscript v8)?**
>
> The $v$ is the **characteristic function** or **characteristic map** in the literature of coalition game theory. Please refer to the following references. The definition of $v$ depends how we define a coalition game. We define a '**spectral coalition game**' in Section 4.2 "Spectral coalition game". We show the implementation of $v$ in Section 4.4 "**Spectral coalition filtering**" and 4.5 "**Characteristic function design and characterization**". Defining the $v$ needs much efforts to prepare.  For example, we need to implement coalitions at coding level defined in the spectral coalition game. This is why the definition of $v$ is so late as it depends on much preparations and preliminaries.
>
> In the Section 4.5 "**Characteristic function design and characterization**", we have defined the **implementation** of the $v$ in terms of our problem by:
> $$
> v(\widetilde{\mathcal{I}}; Q, \mathcal{D}) := \mathop{\mathbb{E}}_{x,y \thicksim (\mathcal{X}, \mathcal{Y})}
> \left\\{ \log Q(y | x \star \widetilde{\mathcal{I}}) - \log Q(y | x \star \emptyset)\right\\}
> $$
> in which the operator $\star$ is defined in Section 4.4 "** Spectral coalition filtering **". This is because the definition of characteristic function or characteristic map $v$ is only an abstract definition. It needs much efforts. For example, how we implement the "coalition" at **physical** level.
>
> Intuitively, the $v$ in IASIDE measures the performance expectation over a dataset. We will show this measure is an estimation of the mutual information between the representations and labels. We have conducted theoretical analysis in Section 4.5 "**Characteristic function design and characterization
> **"
>
> References:
> - [1]. Roth, Alvin E., ed. The Shapley value: essays in honor of Lloyd S. Shapley. Cambridge University Press, 1988.
> - [2]. Roth, Alvin E. "Introduction to the Shapley value." The Shapley value (1988): 1-27.
> - [3]. Lectures on Cooperative Game Theory, https://personal.utdallas.edu/~chandra/documents/6311/coopgames.pdf
>
>
> ## **C3: In eq. 3, you used the convolution of $x$ with $\widetilde{I}$, then how is this computed for $\widetilde{I} \cup \{\mathcal{I}_i\}$?**
>
> Thanks for raising this concern.
>
> In the manuscript, we define:
> $$
> \widetilde{x}_{\widetilde{\mathcal{I}}} =
>     x \star \widetilde{\mathcal{I}} =
> \mathscr{F}^{-1}\left[
> \mathscr{F}(x) \odot  \mathbb{H}(\widetilde{\mathcal{I}}) -
> b \odot (\mathbb{1} - \mathbb{H}({\widetilde{\mathcal{I}}}))
> \right ]
> $$
> where $\star$ is an operator representing the RHS of this equation. It is not a convolution operator.
>
> Please refer to Section 4.4 "**Spectral coalition filtering**". We have revised the section for more general audiences. The readability should be improved.

---

> ### Author Response · Authors · 2024-01-05
> **The revisions and rebuttals as per the requests from the reviewer LUjU (part 2)**
>
> ## **C4: The definition of $Q$ in eq. 5 shows that it is an optimal classifier for a given model and dataset.**
>
>
> We appreciate the concern. We have revised the manuscript and added Section 4.1 "**Notations and preliminaries**". Please also refer to the revised manuscript as per your requests in Section 4.5 **Characteristic function design and characterization**
>
> **IASIDE** does not need to train classifiers. However, we theoretically discuss the training dynamics of general classifiers from the perspective of information theory. We show that:
> - A pre-trained classifier with cross-entropy loss is indeed in maximising a variational mutual information bound (B.A. bound). Please refer to the Lemma 4.1 **B.A. Feature Representation Variational Mutual Information Bound.**
> - The characteristic map $v$ thus is meaningful as it indeed measures the mutual information between the representations of input features and labels for fixed model and dataset.
>
> The reason we use much ink on the simple intuitive $v$ is to conduct a theoretical analysis and show that the choice is very meaningful in terms of information theory.
>
> ## **C5: When you use a perturbed dataset $Q(\mathcal{X} \star \widetilde{I})$ in eq. 8, are you considering another optimal classifier for the perturbed dataset?**
>
> We appreciate your concern. We have added a Section 4.1 "**Notations and preliminaries**" to address the concerns regarding notations.
>
> The notations such as  $Q(\mathcal{X})$ and $Q(\mathcal{X} \star \widetilde{I})$ (also other notations in this manuscript) are adopted from pure math in particular from **functional analysis"" and **topology**. Please refer to the revised manuscript.
>
> For example, let $Q(x): x \mapsto [0,1]^{|\mathcal{Y}|}$ be a classifier where $|\mathcal{Y}|$ denotes the cardinality of labels i.e. the number of classes. The $Q(x)$ can be viewed as the outputs after Softmax when we feed input $x$. The model $Q$ will output $|\mathcal{Y}|$ predictions for each class.
>
> For example, suppose the classifier is a classifier with 10 classes, the cardinality of labels $\mathcal{Y}$ is 10 (i.e. $|\mathcal{Y}|=10$).
>
> In topology or analysis, we often define:
> $$
> Q(\mathcal{X})
>  \stackrel{def}{=}
> \\\{
> Q(x) | x \in \mathcal{X}
> \\\}.
> $$
>
> Thus:
> $$
> Q(\mathcal{X} \star \widetilde{\mathcal{I}})
>  \stackrel{def}{=}
> \\\{
> Q(x) | x \in \mathcal{X} \star \widetilde{\mathcal{I}}
> \\\}
> $$
> where $ \mathcal{X} \star \widetilde{\mathcal{I}}$ denotes the operator $\star$ at set $\mathcal{X}$. Please refer:
> - Section 4.1 Notations and preliminaries
> - Section 4.4 Spectral coalition filtering
>
> We use the notations from rigorous math to formulate the problem. However, they might be dense for general audiences. We have clarified these notations in Section 4.1 "**Notations and preliminaries**" as per your requests.
>
> We can also refer to the following lectures for example:
> - [1]. Metric spaces, https://ocw.mit.edu/courses/18-s190-introduction-to-metric-spaces-january-iap-2023/
> - [2]. Intro to topology, https://math.mit.edu/~jhirsh/topology.html
> - [3]. Functional analysis, https://ocw.mit.edu/courses/18-102-introduction-to-functional-analysis-spring-2021/
>
> The Q in equations is some trained classifier. We do not train any classifier. We just use the equations to characterise and measure the importance with the presence and absence of some frequency components in the dataset.
>
>
> ## **C6: The notation $Q(\mathcal{X})$ is rather confusing as it is called the representations, then it becomes the optimal classifier.**
> We appreciate your concern. We have revised our manuscript and clarified the notations as per your requests. Please refer to our replies regarding C5. Please also refer to:
> - Section 4.1 Notations and preliminaries
> - Section 4.5 Characteristic function design and characterization
>
> The notation $Q(\mathcal{X})$ is a collection of the transformed inputs by the classifier $Q$:
> $$
> Q(\mathcal{X})
>  \stackrel{def}{=}
> \\\{
> Q(x) | x \in \mathcal{X}
> \\\}.
> $$
>
> The transformed representations of inputs are called representations or feature representations. We thank your concern and have clarified regarding features. Please refer to the highlighted part in Section 1:
> > The term use of 'feature' is ambiguous and nuanced in deep learning community. In this research, we clarify and differ the term into: (1) Input features (pixel features) and the representations of input features (feature representations). The pixels of images are referred as 'input features' or 'pixel features'. The 'input features' are fed into image models and the models represent the features inside. The way that models represent the input features is referred as 'feature representations'. Empirically, we can `observe' how models represent the input features by looking into the outputted probability distributions.} Feature representations are the representations of input features that models learn to represent relevant information while neglect irrelevant information ...

---

> ### Author Response · Authors · 2024-01-05
> **The revisions and rebuttals as per the requests from the reviewer LUjU (part 3)**
>
> ## **C7: It is not clear how can $Q$ be related to define a classifier with random weights in eq. 13.**
>
> We appreciate your concern. We have revised the manuscript as per your requests regarding this. Please refer to:
> - Section 4.1 "**Notations and preliminaries**" regarding how Q is defined as an image classifier.
>
> The pytorch code for **randomising** a classifier $Q$ is:
> ```python
> for n, p in Q.named_parameters():
>     nn.init.normal_(p)
> ```
>
> ## **C8: What is the definition of $\widetilde{v}$ ?**
>
> We appreciate your concern. To address the concern regarding $v$. We have revised the manuscript by clarifying: The map $v$ is defined on the top of:
> - (1). The spectral coalition $\widetilde{\mathcal{I}} \in 2^\mathcal{I}$,
> - (2). The classifier $Q$,
> - (3). The dataset $\mathcal{D}:= (\mathcal{X}, \mathcal{Y})$.
>
> That is $v(\widetilde{\mathcal{I}}; Q, \mathcal{D})$. To maintain the depiction succinct, we denote $v(\widetilde{\mathcal{I}}; Q, \mathcal{D})$ as $v(\widetilde{\mathcal{I}})$.
>
> In the equation, $\widetilde{v}$ means one randomised classifier by **randomising** the weights of $Q$. In terms of computing expectations, the randomisation can take place multiple times to derive the expectations.
>
> Please refer to:
> > ... where $\widetilde{v}$ denotes the characteristic function of some model with randomly initialized weights and $\mathbb{E}_{\widetilde{v}} [{\beta}^T \Psi^*(\widetilde{v})]$ denotes the statistical expectation of the measured distributions when the model over all random weights. ...
>
> in the second paragraph of the Section 4.6 **Summarizing spectral importance distribution**
>
> ## **C9: Why it is $\frac{1}{M}$ in eq. 14?**
>
> Thanks for the concern. In the **motivation** experiments in the **second row** of Figure 4 of manuscript v9, we have empirically observed that the models with **randomised** weights **DO NOT** exhibit the preference in low-frequency. The measured spectral importance distributions are **uniform**.
>
> We refer the **random decisions** as the weights of classifiers are **randomised**. **Random decision** can be used as baseline to benchmark how good classifiers will be in making decisions. Thus, using the results from the motivation experiments in Figure 4, we can **assume** the **ideal** **random decisions** should be ideal **uniform**. This means the spectral importance should be $\frac{1}{M}$:
> $$
> \mathbb{E}_{\widetilde{v}} [{\beta}^T \Psi^*(\widetilde{v})] \approx {\beta}^T \frac{\mathbb{1}}{M}.
> $$
>
>
> ## **C10: Introduction: when you talk about how features are represented inside black box models. What do the features mean? This point is not so clear from the rest of the article.**
>
> We sincerely appreciate your suggestion! We have revised our manuscript as per your request. Please refer to:
> > The term use of 'feature' is ambiguous and nuanced in deep learning community. In this research, we clarify and differ the term into: (1) Input features (pixel features) and the representations of input features (feature representations). The pixels of images are referred as 'input features' or 'pixel features'. The 'input features' are fed into image models and the models represent the features inside. The way that models represent the input features is referred as 'feature representations'. Empirically, we can `observe' how models represent the input features by looking into the outputted probability distributions.} Feature representations are the representations of input features that models learn to represent relevant information while neglect irrelevant information ...
>
> in the Section 1.

---

> ### Author Response · Authors · 2024-01-05
> **The revisions and rebuttals as per the requests from the reviewer LUjU (part 4)**
>
> ## **C11: The authors mentioned in the introduction that interpreting the global feature important in spatial domain has inherited limits. Is it possible to illustrate some of these limitations in your applications, to compare with your proposed method?**
>
> We appreciate your concern. We have revised the manuscript as per your requests. Please refer to the "**Global interpretability**" in Section 3 Related work:
> > ...
> Interpreting the global feature importance in the spatial domain for images has inherited limits. For example, the showcase experiment of literature \citep{covert2020understanding} provides the global pixel-wise explanation of a multi-layer perceptron (MLP) with the MNIST on spatial domain. The method assigns each pixel with an importance score and adds the scores over MNIST to form the global interpretation. However, the positions and angles of the objects in images can vary from image to image. Adding the pixel importance scores as global interpretation only offers very limited information. We only read that models pay attentions in the centers of the images in MNIST \citep{covert2020understanding}.} Thus the interpretations in spatial domain often suffer from these limits and provide very limited insights regarding robustness. **Our work differs from these prior works in that we interpret global model robustness in frequency domain. \textcolor{blue}{Frequency-domain transforms are not sensitive to the positions and angles of the objects of images. For example, the spectrum of rotated or flipped images remains identical. The frequency-invariant property in tandem with the correlation to perturbation robustness allows \methodname~to interpret model decision behaviors globally**
> ...
>
> In particular, please. refer to the experiment Figure 1 of the page 8 in the literature [1].
>
> References:
> - [1]. Ian Covert, Scott M Lundberg, and Su-In Lee. Understanding global feature contributions with additive
> importance measures. Advances in Neural Information Processing Systems, 33:17212–17223, 2020. Link: https://arxiv.org/abs/2004.00668
>
>
>
> ## **C12: The sensitivity analysis regarding the choice of hyper-parameters in the proposed approach can also added. In particular, would this explain why there is an outlier in the score correlation for the shufflenet in Fig 10? What happens if the eps is not set to 0.1.**
>
> We have tested it in v7. But in v8 we haven't updated the supplementary materials yet. We have tested multiple eps. Our results still hold.
>
> Please refer to the Appendix: **Full experiments regarding adversarial perturbations** in page 28 of the revised manuscript v9.
>
>
> ## **C13: How the errors in Figure 12 are related to the eps in eq. 20?**
>
> The error bound $\epsilon_K$ in theoretical analysis is an absolute bound with respect to the number of samples $K$:
> $$
> \epsilon_K
>      \stackrel{def}{=} \sup_{i} \mathbb{E}_{(\mathcal{X}, \mathcal{Y})}
>  ||\psi_i^*(\mathcal{I},v) - \psi_i(\mathcal{I},v)||_1
>  \leq  2^{M-1} \cdot \left\\\{ \frac{Var(\Delta v^*)}{K} \right\\\}^{\frac{1}{2}}.
> $$
>
> Clearly, $\lim \epsilon_K \mapsto 0$ as $K \mapsto \infty$. To show this statement hold, we can show the relative errors converge to zero. This is can be understand as: The measuring approach in our method is Cauchy sequence. Each Cauchy sequence converges. The converging implies the correctness the error bound.
>
> We use the relative errors in experiments, they are easy to visualise for understanding. This is also because we have no knowledge what is the **limit** of the measuring sequences. But we know series converge to some limit.
>
>
>
> ## **C14: Given the existing literature, it is still not very clear what is the main advantage of using this method, besides the claim that the proposed method seems to be a quite general tool. The connection between the proposed score and the intrinsic property of the studied models would require further study.**
>
> We agree with you. **We mainly and simply aim to propose a method of measuring the spectral importance.** The applications in specific scenarios indeed need further investigations in our future work.  Unfortunately, in current manuscript, we have to focus on the method itself. **We appreciate the concern, we will continue the research in future papers by discussing these concerns.**
>
> Please kindly refer to the contribution statement in Section 2 **Contributions.**
>
>
> We sincerely appreciate your efforts and help.
>
> Thank you again.
>
> Authors.

---

> ### Author Response · Authors · 2024-01-06
> **Regarding outliers.**
>
> ## Regarding outliers such as shufflenet...
>
> >  ...In particular, would this explain why there is an outlier in the score correlation for the shufflenet in Fig 10? What happens if the eps is not set to 0.1.
>
> Unfortunately, we do not expect that IASIDE can explain the behaviours for all models. Analysing outliers needs to be done case by case. We have made a statement in our manuscript:
>
> > The outliers remind us: Spectral importance distributions can merely reflect one aspect of the robustness of models.
> of the Section 5.4 **Showcase A4: Understanding adversarial vulnerability** and the Section 6 **Limitations**.
>
> Regarding shufflenet, we can only surmise:
> > ... The \textit{shufflenet} uses two techniques: (1) Point-wise group convolution and (2) channel shuffle. The channel shuffle operations in \textit{shufflenet} encourage the network to . ...
>
> The shuffling operations may have effects of encouraging the learning of robust features but IASIDE can not capture in terms of spectral importance. In our experiments, we found training shufflenet is way difficult than other architectures. The architecture is very sensitive to hyper-parameters. This means the cost is not free.
>
> We removed this content from our manuscript as it is not in our main discussion stream.
>
> We wish to save the investigation for outliers for future papers by answering: **Why some architectures are outliers**.
>
> We thank your efforts and the constructive comments.
>
> Thank you again sincerely.
>
> Authors.

---

> > ### Comment · Reviewer_LUjU · 2024-01-09
> > **minor comments**
> >
> > Dear authors,
> > Thanks for your effort to improve the article quality .
> > I am leaning to accept  the article. Here are some minor comments:
> > - Clarify the computational complexity of the method to compute psi^ast(v) or eq 5, as a function of M
> > - Clarify the M in Fig 3 in v9, which shows that M=8, but there seems only 4 middles images of x*I_k with k=0..3 .
> > - The bound v(tilde I) in eq 16 is not so significant in terms of the final score in eq. 5. To have an interepresentation of  psi^ast(v) or eq 5 would be a plus.
> > - Discuss how to the hyper-param beta in eq 19.
> >
> > best,

---

> > > ### Author Response · Authors · 2024-01-09
> > > **We appreciate your great help in assessing our work and your recommendation.**
> > >
> > > To Reviewer LUjU:
> > >
> > > We sincerely appreciate your great efforts in helping us to improve our research soundness and quality. Our manuscript covers many aspects and details which need great efforts in assessing. We thank you for your time and valuable comments again.
> > >
> > > We are now working on the revisions as per your requests. We will online a revised manuscript shortly.
> > >
> > > Thank you again.
> > >
> > > Authors.

---

### Decision · Action_Editor_KvPR · 2024-01-11

**Recommendation:** Reject

**Comment:**

All reviewers agree that the ideas proposed in this manuscript are novel, and that they could be of interest to a large audience in machine learning, signal processing and computer vision. In particular, the reviewers value the study of global spectral feature importance and their connections to robustness.

At the same time, two reviewers remain concerned on the clarity of the presented ideas, even after revisions. These reviewers argue that the manuscript is at times difficult to comprehend, including several hand-wavy comments and with notation usage that is still unclear. I have personally revised the manuscript and I agree with these comments: while the concepts are interesting and would be worthy of publication, the text is hard to follow, there are several grammatical mistakes, and some of the reviewers' comments on notation are still unresolved.

With this context, I am recommending this paper be rejected. However, I encourage the authors to address all of the comments provided by the reviewers in a major revision, putting emphasis on the clarity of presentation and statements, after which they could re-submit this work to TMLR for consideration.

A.E.

**Audience:**

This work could be of interest to a large audience in TMLR, who tries to understand methods to quantify feature importance and interpretability, as well as to understand aspects of robustness and brittleness of modern deep networks.

**Claims And Evidence:**

This paper has received mix ratings and recommendation from 3 reviewers. While one reviewer is content with the level of evidence for the claims in this paper, two reviewers show significant doubt on the clarity of the presentation, the preciseness of the statements, and the support for some of the claims.

**Resubmission Of Major Revision:**

The authors may consider submitting a major revision at a later time.

---

> ### Author Response · Authors · 2024-01-15
> **To AE and reviewers.**
>
> To AE and reviewers,
>
> Although the decision is disappointed , we sincerely thank the constructive comments from AE and all reviewers.  We have spent huge efforts and time to run extra experiments, overhaul and re-organize the manuscript and manage to satisfy all the reviewers in past few months. As a result, the manuscript was increased from 12 pages to 18 pages as per the requests in extra experiments and structures. We still want to say thanks for all reviewers.
>
> We may consider of re-submitting our manuscript after a major revision regarding the clarification.
>
> We thank the great efforts made on our manuscript from AE and reviewers again.
>
> Authors.